# Adaptive Context Length Optimization with Low-Frequency Truncation for Multi-Agent Reinforcement Learning

**Wenchang Duan**[1]**, Yaoliang Yu**[2]**, Jiwan He**[1]**, Yi Shi**[1]*
[1]Shanghai Jiao Tong University     [2]University of Waterloo
{duanwenchang, kih020429, yishi}@sjtu.edu.cn
yaoliang.yu@uwaterloo.ca

## Abstract

Recently, deep multi-agent reinforcement learning (MARL) has demonstrated promising performance for solving challenging tasks, such as long-term dependencies and non-Markovian environments. Its success is partly attributed to conditioning policies on large fixed context length. However, such large fixed context lengths may lead to limited exploration efficiency and redundant information. In this paper, we propose a novel MARL framework to obtain adaptive and effective contextual information. Specifically, we design a central agent that dynamically optimizes context length via temporal gradient analysis, enhancing exploration to facilitate convergence to global optima in MARL. Furthermore, to enhance the adaptive optimization capability of the context length, we present an efficient input representation for the central agent, which effectively filters redundant information. By leveraging a Fourier-based low-frequency truncation method, we extract global temporal trends across decentralized agents, providing an effective and efficient representation of the MARL environment. Extensive experiments demonstrate that the proposed method achieves state-of-the-art (SOTA) performance on long-term dependency tasks, including PettingZoo, MiniGrid, Google Research Football (GRF), and StarCraft Multi-Agent Challenge v2 (SMACv2).

## 1 Introduction

Multi-agent reinforcement learning (MARL) has drawn increasing interest in recent years, which provides a promise for facing many complex real-world challenging problems such as transportation management [1], robot control [2], and finance [3]. However, due to the long-term dependencies and non-rigorous Markovianity of complex tasks, contextual information is introduced to assist policy making [4, 5, 6]. Accordingly, this places a substantial demand on how to leverage contextual information and to what extent [7, 8].

Existing methods are mostly applied to single-agent reinforcement learning (RL), where contextual information performs reasonably well in simple tasks [9, 10, 11]. In comparison, multi-agent reinforcement learning (MARL) involves significantly more complex tasks [12, 13], where relying solely on short context lengths or individual observations often results in suboptimal performance. To address this, one natural approach is to extend the context length. However, the expansion of the context length leads to two significant challenges: the first is an increase in necessary computation, and the second is the difficulty of high dimensionality of the input representation and generalization [14].

---

*Corresponding author: Yi Shi (yishi@sjtu.edu.cn).
†Project code is available at: https://github.com/duanwenchang/ACL-LFT.

39th Conference on Neural Information Processing Systems (NeurIPS 2025).

To address the challenge of increasing computation, references [9, 15] optimized the needed context length and the utilization efficiency; references [16, 17] adopted parallel computation; and reference [18] enhanced the performance of modern hardware. However, the above methods involve a long-time pre-training process, and eventually only obtain static context length. These static context lengths are difficult to adapt to changing environments, which potentially leads to suboptimal solutions or inefficient use of computational resources.

The challenge of input representation and generalization remains unresolved in the field of multi-agent reinforcement learning (MARL) [14]. While recent improvements in processing long sequences that came with attention models significantly alleviate requirements for generalization, an effective representation is still crucial for obtaining optimal contextual information [19, 20, 21]. Regarding the fields where the contextual information is also significant, natural language processing (NLP) typically involves leveraging large language models (LLMs) to autonomously learn and generate prompts, which does not align well with the principles of MARL [22, 23]. Therefore, a tailored input representation for MARL is required.

According to the aforementioned analysis, we propose an adaptive context length optimization with low-frequency truncation (ACL-LFT) for MARL. Specifically, a senior central agent is introduced to adaptively optimize context length, and a tailored attention-based reward is designed to align with the central agent. Via real-time interacting with environment, the central agent determines the optimal context length to address the challenge of increasing computation. Besides, we apply the Fourier transform to map the data from the time domain to the frequency domain, facilitating more effective redundancy filtering compared to direct processing in the time domain. Via truncating the low-frequency band, we obtain an effective input representation for the central agent, which captures the global temporal trends from the decentralized agents. With the above designs, our method effectively solves the dual challenges of increasing context length, achieving efficient leverage of the contextual information.

We benchmark the proposed method across various environments including Sample Spread in PettingZoo [24]; MiniGrid Soccer Game in OpenAI Gym [25]; Academy 3 vs 1 with Keeper, and Academy Counterattack-Hard in Google Research Football (GRF) [26]; *3s5z_vs_3s6z, 5m_vs_6m*, and *corridor* in StarCraft Multi-Agent Challenge v2 (SMACv2) environments [27]. Combined with several types of experiments, including state-of-the-art (SOTA) sequence processing algorithms and different fixed-length methods, we show that the proposed method significantly enhances the performance of the baseline algorithm in changing environments. The main contributions of this paper are summarized as follows:

- To the best of our knowledge, ACL-LFT is the first framework to systematically address the dual challenges of increasing context length in MARL. Equipped with the central agent, our framework achieves adaptive and efficient leverage of contextual information to enhance the decision-making of decentralized agents. Additionally, we present a theorem to theoretically demonstrate the superior performance of adaptive context length over static ones in dynamic environments.

- We propose a novel Fourier-based low-frequency truncation to obtain the global temporal trends from context, effectively addressing the challenge of representing the MARL environment and providing an efficient input for the central agent.

- We empirically demonstrate that the proposed method outperforms SOTA sequence processing algorithms across various long-term dependency environments. We also provide experimental results to demonstrate the superior performance of the proposed method over different fixed lengths in dynamic environments.

## 2 Preliminaries

### 2.1 Decentralized Partially Observable Markov Decision Process with Historical Information

The Decentralized Partially Observable Markov Decision Process with historical information is defined as a tuple $\mathcal{M} = (N, S, A, P, R, \gamma)$, where $N$ is the set of $n$ agents, $S$ denotes the global state space, and $A$ represents the joint action space. At time $t$, the environment evolves according to the transition function $P(s'_t|s_t, a_t)$, which specifies the probability of reaching the next state $s'_t$ given the current global state $s_t = \{s_t^1, s_t^2, \cdots, s_t^n\}, s \in S$ and the joint action $a_t = \{a_t^1, a_t^2, \cdots, a_t^n\}$. The

environment then produces a global reward $r_t = R(s_t, a_t)$. In this decentralized setting, each agent follows a local policy $\pi$ that seeks to maximize the expected cumulative discounted reward, given by $J(\pi) = \mathbb{E}\left[\sum_{t=0}^{\infty} \gamma^t R(s_t, a_t) \,\middle|\, \pi\right]$, where $\gamma \in [0, 1)$ is a discount factor that balances the importance of immediate versus future rewards. The classical Markov property assumes that state transitions depend only on the current state and action.

However, in decentralized partially observable environments, agents cannot directly access the full global state $s_t$; instead, they rely on local observations that are incomplete and noisy. In many scenarios, the assumption that decision-making can be based solely on the current observation is insufficient. To address this, the framework incorporates **historical information** to approximate the underlying dynamics and capture long-term dependencies. Formally, the extended model can be written as $\widetilde{\mathcal{M}} = (N, \widetilde{S}, \widetilde{A}, \widetilde{P}, \widetilde{R}, \gamma)$, where $\widetilde{S}$ includes not only the current state $S$ but also a contextual information space $S^{-1}$ representing observation histories, and $\widetilde{A} = A$. At time $t$, the transition is expressed as

$$P(\widetilde{s_t'}|\widetilde{s}_t, \widetilde{a}_t) = P\big(\widetilde{s}' = s_t' \cup s_t'^{,-1} \,\big|\, \widetilde{s}_t = s_t \cup s_t^{-1}, \widetilde{a}_t = a_t\big),$$

and the reward is given by

$$R(\widetilde{s}_t, \widetilde{a}_t) = R(\widetilde{s}_t = s_t \cup s_t^{-1}, \widetilde{a}_t = a_t).$$

Unlike the standard Markov Decision Process, this extended Decentralized Partially Observable framework enables modeling of complex dynamic environments with long-term temporal dependencies, where leveraging historical information is essential for effective coordination among agents.

## 2.2 The Fourier Transform and Littlewood–Paley Theory

The Fourier transform provides a fundamental tool for analyzing functions in the frequency domain by decomposing signals into their constituent frequency components. Formally, for a function $f \in L^1(\mathbb{R}^d)$, the Fourier transform is defined as:

$$\mathcal{F}f(\xi) = \hat{f}(\xi) = \int_{\mathbb{R}^d} e^{-i(x|\xi)} f(x) \, dx, \tag{1}$$

where $(x|\xi)$ denotes the inner product in $\mathbb{R}^d$. As a continuous linear map from $L^1(\mathbb{R}^d)$ into $L^\infty(\mathbb{R}^d)$, it satisfies $|\hat{f}(\xi)| \leq \|f\|_{L^1}$, ensuring boundedness in the transformed domain. Besides, for any function $\varphi \in L^1$ and an automorphism $L$ on $\mathbb{R}^d$, the transformation obeys:

$$\mathcal{F}(\varphi \circ L) = \frac{1}{|\det L|} \hat{\varphi} \circ L^{-1}. \tag{2}$$

By mapping state representations from the time domain to the frequency domain, the Fourier transform captures underlying structural patterns, where low-frequency components effectively encode global trends while filtering high-frequency noise [28][29].

Littlewood–Paley theory provides a decomposition that functions or distributions are easier to deal with if split into countable sums of smooth functions whose Fourier transforms are compactly supported in a ball or an annulus [30]. In the complex non-Markovianity environments, such decomposition renders a localization procedure in frequency space, which the derivatives act almost as homotheties on distributions. This property establishes fundamental bounds on the behavior of derivatives in different $L^p$ spaces, leading to the following Bernstein inequalities. Let $\mathcal{C}$ be an annulus and $B$ a ball. There exists a constant $C$ such that for any nonnegative integer $k$, any pair $(p, q) \in [1, \infty]^2$ with $q \geq p \geq 1$, and any function $u \in L^p$, the following holds:

$$Supp \quad \hat{u} \subset \lambda B \Rightarrow \|D^k u\|_{L^q} \stackrel{\text{def}}{=} \sup_{|\alpha|=k} \|\partial^\alpha u\|_{L^q} \leq C^{k+1} \lambda^{k+d(\frac{1}{p}-\frac{1}{q})} \|u\|_{L^p}, \tag{3}$$

$$Supp \quad \hat{u} \subset \lambda \mathcal{C} \Rightarrow C^{-k-1} \lambda^k \|u\|_{L^p} \leq \|D^k u\|_{L^p} \leq C^{k+1} \lambda^k \|u\|_{L^p}. \tag{4}$$

The above inequalities highlight a key property: if a function's Fourier spectrum is restricted within frequency $\delta$, its $\alpha$-th order derivative amplifies high-frequency components by a factor of $\delta^{|\alpha|}$. This property enables an efficient truncation of low-frequency information, which serves as an effective representation of contextual information while preserving stability in the decision-making process.

# 3 Methodology

In this section, we propose a novel MARL framework for obtaining adaptive and effective contextual information, which systematically tackles the dual challenges of increasing context length in MARL. The overall framework is shown in Fig. 1, which is comprised of three main components: (1) the Fourier-based low-frequency truncation module, (2) a central agent of adaptive select contextual information, and (3) the structure of learning with spatio-temporal decoupling.

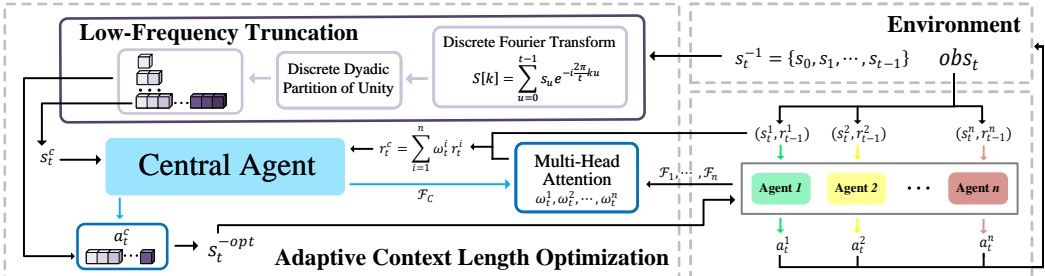

Figure 1: Schematics of our ACL-LFT. At each time $t$, the historical state $s_t^{-1}$ is first processed via the Fourier-based low-frequency truncation module. The central agent leverages the truncated information $s_t^c$ as input and then adaptively optimizes the context length. Subsequently, the decentralized agents then integrate the optimized contextual information $s_t^{-opt}$ with the current state to achieve decision-making.

## 3.1 Fourier-based Low-Frequency Truncation

To achieve an efficient representation of the MARL environment and enhance its effectiveness as input for the central agent, we introduce a low-frequency truncation method. By filtering out high-frequency fluctuations while preserving low-frequency parts, this method captures global temporal trends across decentralized agents and provides a more stable basis for downstream decision-making. Specifically, given the discrete nature of historical state data $s_t^{-1} = \{s_j\}_{j=0}^{t-1}$, we first leverage the Discrete Fourier Transform (DFT) to convert time domain data to frequency domain data:

$$S[k] = \sum_{u=0}^{t-1} s_u e^{\frac{-i2\pi ku}{t}}, \quad k = 0, 1, \ldots, t-1, \tag{5}$$

where $S[k]$ represents the frequency-domain coefficient corresponding to frequency index $k$, while each coefficient encodes a particular oscillatory component of the historical sequence. For real-valued signals, the DFT exhibits conjugate symmetry: $S[k] = S[t-k]^*$, reflecting periodicity in the frequency domain. This transformation effectively disentangles different frequency components of the input sequence, allowing for a more interpretable and structured representation of historical states.

Building upon the Littlewood–Paley theory, we then introduce the Dyadic Partition of Unity method and extend it to the discrete space to truncate the low-frequency information [30]. The Dyadic Partition of Unity method for measurable functions is provided in Appendix A.1. We extend this method to adapt discrete frequency domain historical states. Specifically, we aim for the sum of the window functions to approximate unity across the entire frequency domain:

$$X[k] + \sum_{j=0}^{J-1-m} \Phi_j[k] \approx 1, \quad \forall k = 0, 1, \ldots, N-1, \tag{6}$$

where $X[k]$ is a low-pass window function that retains only the low-frequency components. $\Phi_j[k]$ represents band-pass window functions that separate different frequency bands in a dyadic manner. $J$ is the maximum decomposition level, and for simplicity, we assume that $t = 2^J$. $m$ is a tunable parameter that determines the truncation frequency, ensuring that $2^m < t/2$.

Within this method, the low-frequency region is defined for $k \leq 2^m$ or $k \geq N - 2^m$, where we set $X[k] = 1$ and ensure that $\Phi_j[k] = 0$ for all $j$, thereby preserving the low-frequency information $s_c$.

In the band-pass regions, corresponding to frequency indices satisfying $2^{j+m} \leq k < 2^{j+m+1}$ (or their symmetric counterparts), a single window function $\Phi_j[k]$ is activated with a value of 1, while all other $\Phi_{j'}[k]$ remain zero for $j' \neq j$, ensuring a well-defined partitioning of frequency bands. At transition points, such as $k = 2^{j+m}$, minor gaps may arise due to non-overlapping support. But as the signal length $t$ increases, the gaps become negligible with an error proportionally decreasing as $O(1/t)$, thereby maintaining a stable approximation of equation 6.

The details of Dyadic Partition of Unity in Discrete Form and its rigorous proof are provided in Appendix A.2. By leveraging low-frequency truncation, the above method effectively captures the global temporal trends across decentralized agents, reducing the redundancy of contextual information and serving as an efficient input representation for the subsequent central agent.

### 3.2 The Central Agent of Adaptive Contextual Information Selection

The central agent in our framework serves as a global information processor, adaptively determining the optimal contextual information length for decentralized agents. It is designed to process and analyze only historical information, without directly handling the current state. Specifically, its decision-making process is structured around three key components: state representation, action space, and reward formulation.

Firstly, the state of the central agent is derived through the Fourier-based low-frequency truncation module, which is elaborated in section 3.1. For the discrete historical states $s_t^{-1} = \{s_j\}_{j=0}^{t-1}$ at time $t$, this module extracts the truncated representation $s_t^c$, which effectively represents the global temporal trends across decentralized agents.

Then, the action space of the central agent $A_c$ is defined as the selection of different low-frequency truncation levels, given by:

$$a_t^c \in A_c = \{m_1, m_2, \ldots, m_M\}, \tag{7}$$

where $M$ represents the dimension of $A_c$, and each action $m_i$ corresponds to a different range of preserved low-frequency bands, with each band representing temporal trends with varying degrees of long-term dependency. Given the selected truncation level $m_i$, the corresponding optimal contextual information $s_t^{-opt}$ is obtained by truncation domain.

Finally, to guide its adaptation process, the reward of the central agent is tailored via the multi-head attention mechanism, which weights the influence of decentralized agents. Specifically, the value function estimates and the policy distributions of decentralized agents serve as keys, while the value function estimate and the policy distributions of the central agent serve as the query. We denote the concatenated representation of the value estimate and the policy distribution of agent $i$ as $\mathcal{F}_i$, and that of the central agent as $\mathcal{F}_c$. For each attention head $g$, $\mathcal{F}_i$ and $\mathcal{F}_c$ are projected into the query and key spaces via transformation matrices $W_Q^g$ and $W_K^g$:

$$\mathcal{Q}_c^g = W_Q^g \mathcal{F}_c, \quad \mathcal{K}_i^g = W_K^g \mathcal{F}_i. \tag{8}$$

The attention weight assigned to each decentralized agent is then computed as:

$$\omega_i^g = \frac{\exp\left(\frac{\mathcal{Q}_c^g \cdot (\mathcal{K}_i^g)^T}{\sqrt{d_k}}\right)}{\sum_i \exp\left(\frac{\mathcal{Q}_c^g \cdot (\mathcal{K}_i^g)^T}{\sqrt{d_k}}\right)}, \tag{9}$$

where $d_k$ represents the dimensionality of the key matrix $\mathcal{K}$, ensuring numerical stability. The final attention weight for each agent at time $t$ is obtained by averaging across all heads $\omega_t^i = \frac{1}{head} \sum_{g=1}^{head} \omega_i^g$. At time $t$, for the weights $\{\omega_t^i\}_{i=1}^n$, the reward for the central agent is then derived as a weighted aggregation of the rewards of decentralized agents $r_t^i$:

$$r_t^c = \sum_{i=1}^n \omega_t^i r_t^i. \tag{10}$$

where $\sum_{i=1}^n \omega_t^i = 1$.

Combined with these three components, the parameters of the central agent are updated using gradient-based optimization with advantage estimation. The value function $V(s_t^c)$ is trained to approximate

the expected return through the temporal difference error:

$$\delta_t^c = r_t^c + \gamma V(s_{t+1}^c) - V(s_t^c), \tag{11}$$

where $s_{t+1}^c$ denotes the next state. The policy parameters $\theta$ are then adjusted by maximizing the advantage-weighted objective:

$$\theta \leftarrow \theta + \zeta \nabla_\theta \log \pi(a_t^c|s_t^c)\delta_t^c, \tag{12}$$

while simultaneously minimizing the value function error $\|\delta_t^c\|^2$ through gradient descent.

Building upon the above design, the central agent achieves adaptive optimization of the context length, ensuring that decentralized agents receive the optimal contextual information $s_t^{-opt}$.

Furthermore, to theoretically establish a long-term advantage lower bound of the proposed method over fixed-length methods, we present Theorem 1.

---

**Theorem 1 (Long-Term Advantage Lower Bound of Adaptive Length)** *: At time $t$, let $L_{adap}$ be the adaptive context length, $L_{fix}$ be a fixed context length, and the mutual information loss of $L$ be denoted as $\mathcal{L}_t(L)$. Under mild assumptions , the expected cumulative reward difference between adaptive and fixed context length satisfies the following regret bound:*

$$\sum_{t=1}^{T} (\mathcal{L}_t(L_{fix}) - \mathcal{L}_t(L_{adap})) \geq \Omega(T) - O(T^\alpha) \tag{13}$$

$$= \Omega(T) \quad (\text{when } T \text{ is sufficiently large})$$

*where $0 \leq \alpha < 1$, with $\alpha$ being a non-deterministic parameter whose formal definition is provided in Appendix B.*

---

This theorem demonstrates the long-term advantage of adaptive length policies with the increasingly unstable environment. The result suggests that adaptively adjusting the context length enables more effective information retention over time, leading to significantly lower regret accumulation. The details and proof are provided in Appendix B.

## 3.3 Structure of Learning with Spatio-Temporal Decoupling

In this section, we discuss how to leverage the spatio-temporal decoupling to train the proposed learning framework. Specifically, the training process is divided into two components: the central agent, which is responsible for selecting the optimal contextual information $s_t^{-opt}$; and the decentralized agents, which leverage this information and their current state $s_t$ to optimize their policies. In this framework, the central agent is trained independently to optimize the temporal information component, while the decentralized agents undergo joint training to refine their policies with filtered temporal information and their spatial information. The policy-making and training process of the ACL-LFT algorithm is illustrated in the pseudocode provided in Appendix C.3.

The global optimization objective of the framework is given by the expected sum of discounted rewards over time for decentralized agents:

$$J_i(\pi) = \mathbb{E}\left[\sum_{t=0}^{\infty} \gamma^t R_i(\widetilde{s}_t, \widetilde{a}_t) \mid \pi\right] \tag{14}$$

The central objective is:

$$J_c(\pi) = \sum_{t=0}^{\infty} \gamma^t \sum_{i=1}^{n} \omega_t^i \mathbb{E}[R_i \mid \pi]. \tag{15}$$

By structuring training in this manner, our framework mitigates the challenge of the excessively large parameter search space that typically arises from the joint optimization of both contextual and current information, a factor known to hinder convergence. As a result, our framework not only accelerates the learning process but also ensures that agents can efficiently leverage temporal trends, thereby improving decision-making in complex multi-agent environments.

# 4 Experiments and Analysis

This section evaluates the proposed ACL-LFT framework across various MARL environments. Section 4.1 outlines the experimental setup and baselines. Section 4.2 and Section 4.3 compare ACL-LFT with sequence processing and fixed-length methods. Section 4.4 and Section 4.5 present ablation and case analyses. Finally, Section 4.6 examines the decentralized setting without cross-agent information sharing.

## 4.1 Experiment Setup

**Environments** We consider various tasks, including Sample Spread in PettingZoo [24], MiniGrid Soccer Game in OpenAI Gym [25], Academy 3 vs 1 with Keeper, and Academy Counterattack-Hard in Google Research Football (GRF) [26]. The overview of environments is shown in Fig. 2, while the details of environments and their reward design are provided in Appendix C.1. All experiments are implemented based on the Multi-Agent Proximal Policy Optimization (MAPPO) algorithm [31]. Furthermore, to verify the effectiveness of our proposed method under both complex and large-scale scenarios, as well as to analyze the impact of removing the MAPPO backbone, we conduct additional experiments based on MAPPO, QMIX [32], and QPLEX [33] in the StarCraft Multi-Agent Challenge v2 (SMACv2) environments [27], including *3s5z_vs_3s6z*, *5m_vs_6m*, and *corridor*. The detailed experimental results are presented in Appendix C.4.

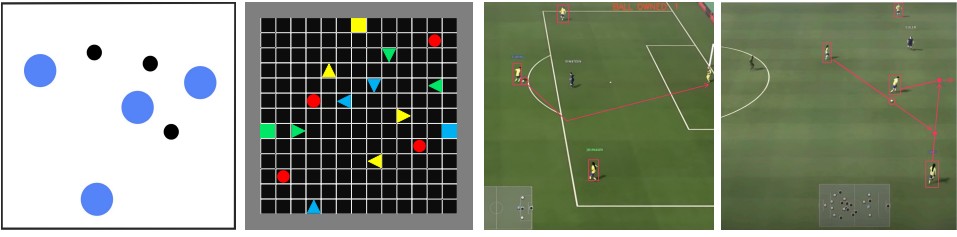

(a) Sample Spread    (b) Minigrid Soccer Game (c) 3 vs 1 with Keeper    (d) Counterattack-Hard

Figure 2: Sample Spread (a) is a search game where agents learn to cover all the landmarks while avoiding collisions. Minigrid Soccer Game (b) is a 15×15 environment where agents (triangles) earn rewards by kicking the ball (circle) into same-colored goalmouths (squares). Academy 3 vs 1 with Keeper (c) is a scenario where three offensive agents attempt to score against one defender and a goalkeeper. Academy Counterattack-Hard (d) is a scenario where four agents must execute a rapid counterattack while avoiding defenders.

Given the varying maximum episode lengths across different environments, we adapt the context length accordingly for each case. Specifically, the maximum episode steps for Sample Spread, MiniGrid Soccer Game, Academy 3 vs 1 with Keeper, and Academy Counterattack-Hard are 25, 512, 400, and 400, respectively. Therefore, the corresponding context lengths are set to 4, 64, 64, and 64 steps. These values define the maximum selectable context lengths for the proposed method, consistent with the central agent's input dimension. Further details are provided in Appendix C.2.

**Baselines** We first benchmark the sequence processing algorithms, including Transformer [34], Token Statistics Transformer (ToST) [35], and AMAGO [36]. The introduction of these methods is provided in Appendix C.2. Then, we benchmark the proposed method against different fixed context lengths.

## 4.2 Performance Comparison with Sequence Processing Methods

In this section, we benchmark the proposed method against Transformer, ToST, and AMAGO. As shown in Fig. 3, the performances are depicted via data from every 100 episodes and averaged over 5 seeds. It is seen that the proposed method outperforms in all scenarios. Specifically, Transformer and ToST need more time to explore sophisticated policies and demonstrate large oscillations in the exploration process. AMAGO demonstrates strong performance during the policy exploration phase, owing to its effective handling of long sequences in parallel. However, due to the fact that its contextual information is of fixed length, which tends to have a lot of noise, it performs poorly compared to the proposed method after convergence.

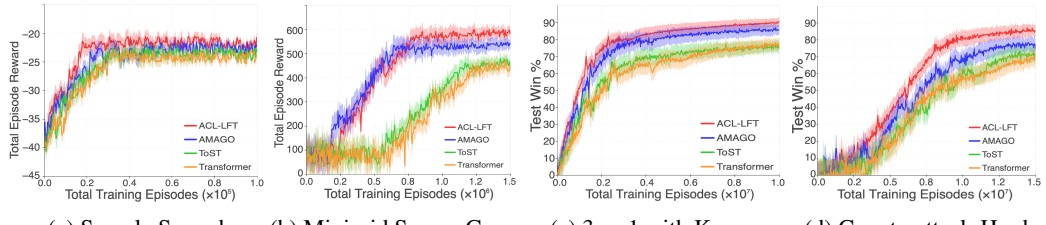

| (a) Sample Spread | (b) Minigrid Soccer Game | (c) 3 vs 1 with Keeper | (d) Counterattack-Hard |

Figure 3: Performance Comparison with Sequence Processing Methods in Four Environments

In the Sample Spread, Academy 3 vs 1 with Keeper and Academy Counterattack-Hard, the proposed method demonstrates the fastest exploration efficiency and consistently achieves the highest post-convergence performance. Notably, as the complexity of the scenarios increases—from Sample Spread to Academy 3 vs 1 with Keeper and Academy Counterattack-Hard—the performance gap between the proposed method and the baseline methods becomes more evident. In the Minigrid Soccer Game, the proposed method and AMAGO exhibit comparable exploration efficiency; however, AMAGO converges prematurely and fails to achieve strong final results. In contrast, the proposed method utilizes low-frequency truncation of historical information, significantly mitigating the impact of redundant data. By adaptively selecting the optimal context length, the proposed method achieves superior performance across all environments.

## 4.3 Performance Comparison with Fixed-Length

In section 3.2, we presented the Theorem 1, which demonstrates the long-term advantage of adaptive length policies over fixed-length. In this section, we benchmark the proposed method against different fixed context lengths (8, 16, 32, and 64 steps) in Academy 3 vs 1 with Keeper and Academy Counterattack-Hard.

We computed the average performance of the proposed method and four fixed-length methods during two relatively stable periods after convergence. Specifically, the results were averaged from the 0.9 millionth to the 1 millionth episode in the Academy 3 vs 1 with Keeper, and from the 1.4 millionth to the 1.5 millionth episode in the Academy Counterattack-Hard. As shown in Fig. 4, the proposed method significantly outperforms all fixed-length methods, further demonstrating the effectiveness and efficiency of adaptive context length optimization. Notably, among the fixed-length methods, the 16-step and 8-step show the best performance in Academy 3 vs 1 with Keeper and Academy Counterattack-Hard, respectively. These findings suggest that longer context lengths do not necessarily lead to better performance, as excessive historical information may introduce significant noise. This observation further underscores the critical role of our low-frequency truncation in enhancing overall performance.

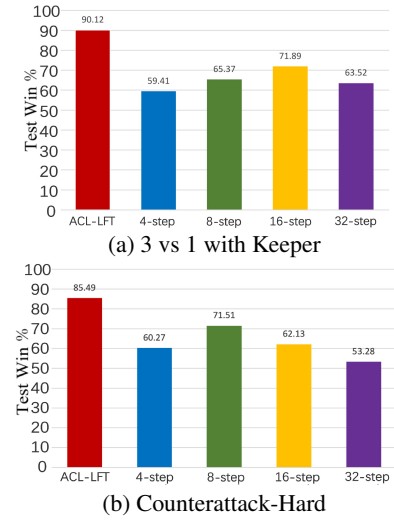

(a) 3 vs 1 with Keeper

(b) Counterattack-Hard

Figure 4: Performance Comparison with Different Fixed Lengths on GRF

## 4.4 Ablation Experiments

To understand the contribution of each component in the proposed ACL-LFT framework, we carry out ablation studies to test the contribution of adaptive context length (ACL) and low-frequency truncation (LFT). Specifically, we utilize the best-performing fixed-length configurations in the Academy 3 vs 1 with Keeper (32-step) and Academy Counterattack-Hard (16-step) environments. These configurations are applied to evaluate ACL-LFT-NO-ACL and ACL-LFT-Raw, which test the performance of the methods without ACL and without both ACL and LFT, respectively. Furthermore, the ACL-LFT-NO-LFT is input with 32 and 32 steps, which align with the maximum step that can be selected by the ACL-LFT.

As shown in Fig. 5, the most impact on performance is the ablation of ACL, which further demonstrates the significant effect of adaptive context length optimizing. Additionally, the results reveal that ACL-LFT-NO-ACL outperforms ACL-LFT-Raw, indicating that low-frequency truncation (LFT) contributes notably to the overall performance of the proposed framework. In conclusion, the results highlight the critical importance of both ACL and LFT in enhancing the efficiency and effectiveness of the overall framework. Moreover, they demonstrate these two components complement each other, contributing synergistically to performance improvement, especially in environments with varying complexities and temporal dynamics.

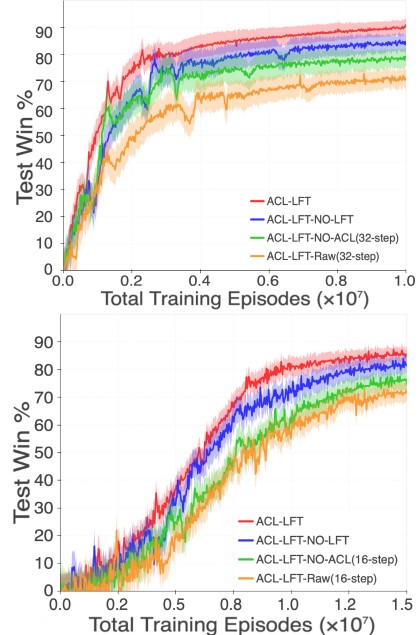

Figure 5: Comparison of ablation studies: (a) 3 vs 1 with Keeper; (b) Counterattack-Hard.

### 4.5 Case Study

To intuitively demonstrate the performance benefits brought by the adaptive context length mechanism, we conduct a case study on the MiniGrid Soccer Game. This environment features a tailored reward function, which is highly sensitive to the agent's ability to leverage historical information for effective path planning and cooperative strategies. Specifically, we set the maximum step of environment to 64. Accordingly, all baseline methods use a fixed context length of 16, which aligns with the maximum selectable length for the proposed method.

As shown in Table 1, we record the reward value every 5 steps and highlight the time step at which the first goal is achieved in bold. The numbers in parentheses indicate the context length adaptively selected by ACL-LFT at each time step. The results show that ACL-LFT quickly adjusts its context length after obtaining positive rewards (e.g., step = 15, 20), enabling timely re-planning to avoid inefficient exploration, while other methods remain in the aimless exploration phase. Notably, ACL-LFT dynamically selects shorter yet effective context lengths (e.g., length 2 at step 40), significantly improving path efficiency and ultimately achieving a goal by step

Table 1: Step-Reward Comparison on MiniGrid Soccer Game

| Step | ACL-LFT | Transformer | ToST | AMAGO |
|------|---------|-------------|------|-------|
| 0 | 0.00 (0) | 0.00 | 0.00 | 0.00 |
| 5 | 2.71 (8) | 2.96 | 0.00 | 0.00 |
| 10 | 1.52 (16) | 1.73 | 2.17 | 2.63 |
| 15 | 1.77 (8) | 1.94 | 1.59 | 2.08 |
| 20 | 2.41 (4) | 1.65 | 1.26 | 1.54 |
| 25 | 3.19 (4) | 2.36 | 1.85 | 1.93 |
| 30 | 3.85 (2) | 2.07 | 2.26 | 2.45 |
| 35 | 3.30 (8) | 2.85 | 2.51 | 2.16 |
| 40 | 4.06 (2) | 3.57 | 2.90 | 2.48 |
| **41** | **14.31 (1)** | 3.45 | 2.97 | 2.65 |
| 45 | / | 4.12 | 3.36 | 3.03 |
| **47** | / | **13.98** | 3.52 | 3.41 |
| 50 | / | / | 3.79 | 3.66 |
| 55 | / | / | 4.09 | 3.85 |
| **56** | / | / | **14.25** | 4.02 |
| **59** | / | / | / | **13.62** |

41. In contrast, methods with fixed-length contexts adapt more slowly, require longer to identify viable paths. This indicates that ACL-LFT enhances exploration efficiency and mitigates the impact of redundant information via adaptive context length optimization, thereby achieving superior performance.

### 4.6 Ablation on the Absence of Cross-Agent Historical Information

To further examine the influence of centralized sequence processing and verify that our method does not rely on cross-agent information sharing, we conduct an additional ablation study in which each agent can only access its own local historical observations and actions, without any global or inter-

agent historical information. In this variant, the central agent is disabled from aggregating histories across agents; instead, it independently processes the local sequences for each agent, ensuring that no centralized communication channel exists during decision-making.

Table 2: Performance comparison without cross-agent historical information.

| Task | AMAGO | Mamba | **ACL-LFT** |
|------|-------|-------|---------|
| 3s5z vs 3s6z | $76.1 \pm 2.9$ | $72.6 \pm 3.2$ | $\mathbf{78.9 \pm 2.8}$ |
| 5m_vs_6m | $48.1 \pm 4.0$ | $46.2 \pm 4.5$ | $\mathbf{52.7 \pm 4.2}$ |
| corridor | $74.3 \pm 4.8$ | $69.0 \pm 5.9$ | $\mathbf{77.9 \pm 5.3}$ |

As shown in Table 2, ACL-LFT consistently outperforms existing temporal modeling methods such as AMAGO and Mamba even when no cross-agent historical information is available. This confirms that the performance improvement of ACL-LFT originates from its proposed low-frequency temporal representation and adaptive contextual-length optimization rather than from any implicit inter-agent information sharing. Moreover, this ablation validates that ACL-LFT preserves the partially observable nature of the SMACv2 environment and remains effective under purely decentralized historical settings, demonstrating the soundness and generality of our framework.

## 5    Conclusion

To systematically address the dual challenges of increasing context length in MARL, we propose an adaptive context length optimization with low-frequency truncation (ACL-LFT) for MARL. The proposed method adaptively optimizes context length via a central agent. Equipped with a Fourier-based low-frequency truncation, we address the challenge of representing the MARL environment and provide an efficient input for the central agent. The experimental results demonstrate the proposed method significantly enhances the performance of the baseline algorithm in changing environments. In addition, we demonstrate both theoretically and experimentally the long-term advantage of adaptive context length over fixed-length.

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

**Appendix Overview:** In this Appendix we provide important details that could not be included in the main text due to space constraints. First in Appendix A, we provide detailed proofs for the low-frequency truncation in discrete space. Next in Appendix B, we provide the proof of the approximately long-term advantage lower bound of the contextual information over static counterparts in dynamic environments. Finally, in Appendix C we provide additional details about our experiments discussed in the main text.

## A   Proof of Low-Frequency Truncation in Discrete Space

### A.1   Dyadic Partition of Unity

The following construction of a dyadic partition of unity is standard in harmonic analysis; see, for example, [30]. We include it here for completeness.

Let $C$ be the annulus defined as $C = \{\xi \in \mathbb{R}^d \mid \frac{3}{4} \leq |\xi| \leq \frac{8}{3}\}$. There exist measurable functions $\chi$ and $\varphi$, taking values in the interval $[0, 1]$, belonging respectively to $\mathcal{D}(B(0, \frac{4}{3}))$ and $\mathcal{D}(C)$, such that:

$$\forall \xi \in \mathbb{R}^d, \quad \chi(\xi) + \sum_{j \geq 0} \varphi(2^{-j}\xi) = 1, \tag{16}$$

and for all:

$$\forall \xi \in \mathbb{R}^d \setminus \{0\}, \sum_{j \in \mathbb{Z}} \varphi(2^{-j}\xi) = 1. \tag{17}$$

These functions satisfy the disjoint support conditions:

$$|j - j\prime| \geq 2 \Rightarrow \operatorname{Supp} \varphi(2^{-j}\cdot) \cap \operatorname{Supp} \varphi(2^{-j\prime}\cdot) = \emptyset, \tag{18}$$

$$j \geq 1 \Rightarrow \operatorname{Supp} \chi \cap \operatorname{Supp} \varphi(2^{-j}\cdot) = \emptyset. \tag{19}$$

Defining the translated annulus $C$ as $Cdef = B(0, \frac{2}{3}) + C$, we note that $C$ remains an annulus and satisfies:

$$|j - j\prime| \geq 5 \Rightarrow 2^{j\prime}C \cap 2^{j}C = \emptyset. \tag{20}$$

Furthermore, the functions $\chi$ and $\varphi$ satisfy the bounds:

$$\forall \xi \in \mathbb{R}^d, \quad \frac{1}{2} \leq \chi^2(\xi) + \sum_{j \geq 0} \varphi^2(2^{-j}\xi) \leq 1, \tag{21}$$

$$\forall \xi \in \mathbb{R}^d \setminus \{0\}, \quad \sum_{j \in \mathbb{Z}} \varphi^2(2^{-j}\xi) \leq 1. \tag{22}$$

The aforementioned method establishes a smooth dyadic decomposition of frequency space using radial functions $\chi$ and $\varphi$, ensuring a partition of unity while maintaining disjoint support conditions. The construction guarantees that the frequency space is effectively covered while avoiding excessive overlap, making it well-suited for applications in harmonic analysis and function space theory. The inequalities further confirm that the decomposition remains stable, with bounded sums ensuring proper reconstruction properties.

### A.2   Proof of Dyadic Partition of Unity in Discrete Form

To extend this formulation to the discrete setting, we consider a similar approach where the continuous frequency domain is replaced by a discrete grid. In this section, we first supplement the details of some important definitions. Then supply the essential proofs of our method.

To facilitate a dyadic-like decomposition in the discrete frequency domain, we introduce a set of window functions (the low-pass window function and the band-pass window functions) that partition the frequency spectrum into complementary regions. These functions ensure that different frequency components of a signal are captured separately, allowing for a structured analysis of its spectral content. Specifically, the low-pass window function $X[k]$ is defined as:

$$X[k] = \begin{cases} 1, & k \leq 2^m \text{ or } k \geq t - 2^m, \\ 0, & \text{otherwise,} \end{cases} \tag{23}$$

where $m$ is an integer satisfying $0 < m < J$, which determines the cutoff for low-frequency retention. This function effectively captures the low-frequency trends of a signal while discarding high-frequency components.

In contrast, the band-pass window functions $\Phi_j[k]$ isolate specific frequency bands and are defined as:

$$\Phi_j[k] = \begin{cases} 1, & 2^{j+m} \le k < 2^{j+m+1} \text{ or } t - 2^{j+m+1} < k \le t - 2^{j+m}, \\ 0, & \text{otherwise,} \end{cases} \tag{24}$$

for $j = 0, 1, \ldots, J - 1 - m$, each function spans a high-frequency dyadic band. These band-pass functions systematically cover the frequency spectrum, ensuring that different frequency components are separately analyzed while maintaining a structured partitioning. Using these window functions, the signal $s[u]$ can be decomposed into distinct components. The low-frequency component, which encapsulates the long-term trend of the signal is obtained as discrete inverse Fourier transform. And the band-pass components, which capture fluctuations at specific dyadic frequency scales are given by:

$$\Delta_j s[u] = \text{IDFT}(\Phi_j[k] \cdot S[k])[u], \quad j = 0, 1, \ldots, J - 1 - m. \tag{25}$$

By summing these components, the original signal can be approximately reconstructed as:

$$s[u] \approx \Delta_{-1} s[u] + \sum_{j=0}^{J-1-m} \Delta_j s[u]. \tag{26}$$

This decomposition demonstrates that the essential features of the signal are effectively captured across multiple frequency scales, allowing for a detailed analysis of its spectral characteristics.

A key property of our method is that the window functions form an approximate partition of unity in the frequency domain. This ensures the decomposition provides a stable and comprehensive representation of the signal. Specifically, the functions satisfy the relation:

$$X[k] + \sum_{j=0}^{J-1-m} \Phi_j[k] \approx 1, \tag{27}$$

for most frequency indices $k$. This property can be verified by examining different frequency regions.

1. **Low-Frequency Regions:** When the frequency index falls within the low-frequency range, the low-pass function fully retains the frequency components, while all band-pass functions are inactive:

$$X[k] = 1, \quad \Phi_j[k] = 0, \quad k \le 2^m \cup k \ge t - 2^m, \quad \forall j. \tag{28}$$

Consequently, the summation property holds:

$$X[k] + \sum_{j=0}^{J-1-m} \Phi_j[k] = 1 + \sum 0 = 1. \tag{29}$$

This ensures that the low-frequency components are preserved without interference from high-frequency bands.

2. **Band-Pass Regions:** In the band-pass regions, the low-pass function does not contribute, while exactly one band-pass function is active:

$$X[k] = 0, \quad \exists! \, j \text{ s.t. } \Phi_j[k] = 1, \quad 2^{j+m} \le k < 2^{j+m+1}. \tag{30}$$

This guarantees that the sum remains unity:

$$X[k] + \sum_{j=0}^{J-1-m} \Phi_j[k] = 0 + 1 = 1. \tag{31}$$

Thus, each frequency component is assigned uniquely to one of the band-pass filters, ensuring no overlap or redundancy.

3. **Boundary Points:** At the boundary points ($k = 2^{j+m}, 2^{j+m+1}, t - 2^{j+m}, t - 2^{j+m+1}$) where transitions occur between different frequency bands, both the low-pass and band-pass window functions may be inactive, leading to:

$$X[k] = 0, \quad \Phi_j[k] = 0, \quad \forall j. \tag{32}$$

Consequently, at these discrete boundary points, the summation deviates from unity:

$$X[k] + \sum_{j=0}^{J-1-m} \Phi_j[k] = 0. \tag{33}$$

To further analyze the convergence properties of this decomposition, we define the error function:

$$E[k] = 1 - \left( X[k] + \sum_{j=0}^{J-1-m} \Phi_j[k] \right). \tag{34}$$

The set of indices where $E[k] \neq 0$ is given by:

$$\mathcal{E} = \{k \in \mathbb{Z} \mid E[k] \neq 0\}. \tag{35}$$

Since $E[k]$ is nonzero only at a finite number of boundary points, the measure of its support satisfies:

$$|\mathcal{E}| \ll t. \tag{36}$$

Thus, as $t \to \infty$, the fraction of affected indices vanishes, leading to an almost everywhere convergence:

$$\lim_{t \to \infty} \frac{|\mathcal{E}|}{t} = 0. \tag{37}$$

The above proofs ensure that the partitioning scheme provides a stable and asymptotically exact decomposition in the frequency domain.

Another fundamental aspect of our method is the disjointness and independence of the window functions, which guarantees that the extracted components remain distinct and do not interfere with each other. This separation is maintained through the following two properties:

1. **Between different band-pass windows:** For any $j \neq j\prime$, the support intervals of the band-pass window functions are disjoint:

$$[2^{j+m}, 2^{j+m+1}) \cap [2^{j\prime+m}, 2^{j\prime+m+1}) = \emptyset. \tag{38}$$

As a result, the corresponding window functions satisfy:

$$\Phi_j[k] \cdot \Phi_{j\prime}[k] = 0, \quad \forall k. \tag{39}$$

2. **Between the low-pass and band-pass windows:** The low-pass function $X[k]$ is supported in the low-frequency regions $k \leq 2^m$ or $k \geq t - 2^m$. On the other hand, each band-pass function $\Phi_j[k]$ is supported in the range $2^{j+m} \leq k < 2^{j+m+1}$. Since $2^{j+m} > 2^m$, it follows that the support of $X[k]$ and $\Phi_j[k]$ are mutually exclusive, ensuring:

$$X[k] \cdot \Phi_j[k] = 0, \quad \forall j, k.$$

Based on the above method, the annulus $C$ is substituted with a corresponding set in the discrete Fourier domain, and the dyadic scaling operations are adapted to respect the discrete nature of the transform. The goal remains to construct a stable decomposition that preserves the essential properties of the continuous case while accommodating the constraints imposed by discrete sampling.

## B    Proof of Long-Term Advantage Lower Bound of Adaptive Length

**Theorem 1** (Advantage Lower Bound of Adaptive Length): At time $t$, let $L_{\text{adap}}$ be the adaptive context length, $L_{\text{fix}}$ be a fixed context length, and the mutual information loss of $L$ be denoted as

$\mathcal{L}_t(L)$. Under mild assumptions , the expected cumulative reward difference between adaptive and fixed context length satisfies the following regret bound:

$$\sum_{t=1}^{T} (\mathcal{L}_t(L_{\text{fix}}) - \mathcal{L}_t(L_{\text{adap}})) \geq \Omega(T) - O(T^{\alpha}) \tag{40}$$

$$= \Omega(T) \quad \text{(when } T \text{ is sufficiently large)}$$

where $0 \leq \alpha < 1$, with $\alpha$ being a non-deterministic parameter that varies with the environment.

**Proof:** To prove this theorem, we first model the non-stationary environment, including the observation and its latent variable. Specifically, at each time $t$, the agent receives noisy observations $o_t = g(\xi_t) + \epsilon_t$, where $\{\xi_t\}_{t=1}^{T}$ represents the latent variable, and the error term $\epsilon_t \sim \mathcal{N}(0, \sigma_\epsilon^2)$ [37]. The latent variable sequence $\{\xi_t\}_{t=1}^{T}$ governs environmental dynamics and follows a diffusion process [38]:

$$d\xi_t = \mu(\xi_t)dt + \eta(\xi_t)dW_t, \tag{41}$$

where $W_t$ represents a standard Brownian motion, $\mu(\cdot)$ and $\eta(\cdot)$ denote the drift and diffusion coefficients, respectively. This diffusion model should be viewed as a modeling approximation of the discrete-time dynamics (e.g., via an Euler–Maruyama discretization with unit time step), and it is imposed to capture gradual stochastic drift in the latent state.

For any window length $L$, the mutual information $I(a_t; \xi_t|L)$ quantifies the information content of the action $a_t$ about the latent variable $\xi_t$:

$$I(a_t; \xi_t|L) = H(\xi_t) - H(\xi_t|a_t, L), \tag{42}$$

where $H(\xi_t)$ represents the entropy of $\xi_t$ (prior entropy), and $H(\xi_t|a_t, L)$ represents the conditional entropy of $\xi_t$ given $a_t$ and window length $L$. With the theoretical optimal window length $L_t^*$, the mutual information loss of $L$ can be obtained:

$$\mathcal{L}_t(L) = I(a_t; \xi_t|L_t^*) - I(a_t; \xi_t|L) \tag{43}$$

Since the action $a_t$ is generated from the observation sequence $o_{t-L:t} = \{o_{t-L}, \ldots, o_t\}$ via the stochastic policy $\pi(a_t|o_{t-L:t})$, the data processing inequality implies:

$$I(a_t; \xi_t|L) \leq I(o_{t-L:t}; \xi_t|L). \tag{44}$$

With the equation 42, we obtain:

$$I(o_{t-L:t}; \xi_t|L) = H(\xi_t) - H(\xi_t|o_{t-L:t}). \tag{45}$$

Thus, the upper bound on the mutual information loss is given by:

$$\mathcal{L}_t(L) \leq I(o_{t-L:t}; \xi_t|L_t^*) - I(o_{t-L:t}; \xi_t|L) = H(\xi_t|o_{t-L:t}) - H(\xi_t|o_{t-L_t^*:t}). \tag{46}$$

Based on the Fourier-based truncation, we assume the policies can approximately fully utilize the observed information; thereby, we obtain:

$$\mathcal{L}_t(L) \approx H(\xi_t|o_{t-L:t}) - H(\xi_t|o_{t-L_t^*:t}). \tag{47}$$

Next, considering the general situation that the latent variable $\xi_t$ can be regarded as a Gaussian form for the posterior distribution $p(\xi_t|o_{t-L:t})$ [37]. This setting is reasonable under several mild and widely satisfied conditions: (i) the dynamics and observation models can be locally approximated as linear or weakly nonlinear in the neighborhood of the true latent state $L_t^*$ [37][38]; (ii) the observation noise is Gaussian, consistent with the structure of the Kalman filter [37]; and (iii) the functions $\mu(\xi_t)$ and $\eta(\xi_t)$, often parameterized by neural networks, are smooth and differentiable, allowing for local Taylor expansions or moment-based approximations such as sigma-point propagation [37][38]. Therefore, the posterior distribution can be reasonably approximated as:

$$p(\xi_t|o_{t-L:t}) = \mathcal{N}(\hat{\xi}_t, \sigma_{t|L}^2), \tag{48}$$

where $\hat{\xi}_t$ denotes the posterior mean and $\sigma_{t|L}^2$ the posterior variance. Consequently, the conditional entropy can be computed as:

$$H(\xi_t|o_{t-L:t}) = \frac{1}{2}\log(2\pi e \sigma_{t|L}^2). \tag{49}$$

Notably, this Gaussian approximation is particularly justified in policy optimization algorithms such as PPO [39], where updates are constrained to remain close to the current policy, effectively preserving local linearity in the latent-to-observation mapping.

Due to the fact that $L_t^*$ is the optimal window length at time step $t$, we assume that the posterior variance $\sigma_{t|L}^2$ behaves in a manner where it decreases as the window length approaches the optimal value $L_t^*$ [40]. This behavior is common in many estimation problems, where the system's performance improves as it gets closer to the optimal configuration. In particular, reference [41] suggests that the most useful information for decision-making is concentrated around certain "bottleneck" points, and small deviations from the optimal choice result in progressively diminishing returns in terms of information processing. Given this intuition, we assume that the posterior variance $\sigma_{t|L}^2$ can be regarded as behaving convexly in the neighborhood of $L_t^*$, because this ensures that small changes around the optimal window length lead to an increase in variance, which represents a deterioration in the quality of the estimation. Specifically, there exists a positive constant $k > 0$ such that:

$$\sigma_{t|L}^2 \geq \sigma_{\min}^2 + \frac{1}{2}k(L - L_t^*)^2, \tag{50}$$

where $\sigma_{\min}^2 = \sigma_{t|L_t^*}^2$ denotes the posterior variance achieved at the optimal window length.

Combining equations 47 and 49, the mutual information loss can be lower bounded in terms of the posterior variance ratio:

$$\mathcal{L}_t(L) \approx \frac{1}{2}\log\left(\frac{\sigma_{t|L}^2}{\sigma_{\min}^2}\right). \tag{51}$$

Substituting the lower bound in equation 50 into equation 51, we obtain:

$$\mathcal{L}_t(L) \geq \frac{1}{2}\log\left(1 + \frac{k(L - L_t^*)^2}{2\sigma_{\min}^2}\right). \tag{52}$$

Applying the inequality $\log(1 + x) \geq \frac{x}{1+x}$ for $x > -1$, we further derive:

$$\mathcal{L}_t(L) \geq \frac{1}{2} \cdot \frac{\frac{k(L-L_t^*)^2}{2\sigma_{\min}^2}}{1 + \frac{k(L-L_t^*)^2}{2\sigma_{\min}^2}} = \frac{k(L - L_t^*)^2}{4\sigma_{\min}^2 + 2k(L - L_t^*)^2}. \tag{53}$$

For a fixed window length $L_{\text{fix}}$, the total information loss over $T$ time steps can then be bounded from below:

$$\sum_{t=1}^{T}\mathcal{L}_t(L_{\text{fix}}) \geq \sum_{t=1}^{T}\left(\frac{k}{4\sigma_{\min}^2}(L_{\text{fix}} - L_t^*)^2 \cdot \frac{1}{1 + \frac{k}{2\sigma_{\min}^2}(L_{\text{fix}} - L_t^*)^2}\right). \tag{54}$$

When $|L_{\text{fix}} - L_t^*|$ exceeds a threshold $\zeta$, the denominator in equation 53 is bounded, and the loss is lower bounded by a positive constant $c = \frac{k\zeta^2}{4\sigma_{\min}^2 + 2k\zeta^2}$. On the other hand, when $|L_{\text{fix}} - L_t^*| < \zeta$, the quadratic term dominates and the loss scales as $\mathcal{L}_t(L_{\text{fix}}) \geq \frac{k}{4\sigma_{\min}^2}(L_{\text{fix}} - L_t^*)^2$. Therefore, for every $t$, we have the pointwise lower bound $\mathcal{L}_t(L_{\text{fix}}) \geq \min\left\{c, \frac{k}{4\sigma_{\min}^2}(L_{\text{fix}} - L_t^*)^2\right\}$. Then, due to $L_{\text{fix}}$ does not coincide with the time-varying optimum for all but a vanishing fraction of rounds. Therefore, we assume there exist constants $\rho \in (0, 1]$ and $\zeta > 0$ such that $\left|\{t \in \{1, \ldots, T\} : |L_{\text{fix}} - L_t^*| \geq \zeta\}\right| \geq \rho T$, or, more generally, that $\sum_{t=1}^{T}(L_{\text{fix}} - L_t^*)^2 = \Omega(T)$. Under either condition, the above per-step lower bounds imply

$$\sum_{t=1}^{T}\mathcal{L}_t(L_{\text{fix}}) = \Omega(T), \tag{55}$$

demonstrating that any fixed window length incurs linear cumulative loss over time unless it tracks the optimal $L_t^*$ (in the sense of maintaining a non-vanishing tracking error frequency or a linear-order cumulative squared tracking error).

Considering the adaptive strategy for selecting the context length $L_{\text{adap}}$ at each time $t$. We adopt the most universal polynomial rate to describe the adaptive policy converging to the optimal window length [42]:

$$|L_{\text{adap},t} - L_t^*| \leq \epsilon_t = O(t^{-\beta}), \quad \beta > 0. \tag{56}$$

which are commonly used in general convex problems and in stochastic algorithms. In contrast, exponential convergence, etc., often requires extremely strong assumptions such as strong convexity and global smoothness.

Next, based on the equation 53, for small deviations $|L - L_t^*| < \zeta$, applying the first-order approximation $\log(1 + x) \approx x$, we obtain:

$$\mathcal{L}_t(L) \approx \frac{k}{4\sigma_{\min}^2}(L - L_t^*)^2. \tag{57}$$

This shows that in a local region $|L - L_t^*| < \zeta$, the mutual information loss behaves quadratically with respect to $(L - L_t^*)^2$. This local quadratic behavior is compatible with the global Lipschitz condition via the boundedness of gradients and compactness of the parameter space.

Moreover, in reinforcement learning algorithms such as TRPO [43] or PPO [39], policy updates are performed under a trust-region style constraint that limits distributional shift, typically expressed via a KL-divergence threshold (or a clipped surrogate that approximates such a constraint). As a canonical form, TRPO enforces an average KL constraint between successive policies, i.e.,

$$\mathbb{E}_o\left[D_{\mathrm{KL}}\left(\pi_{\theta_t}(\cdot \mid o) \,\|\, \pi_{\theta_{t+1}}(\cdot \mid o)\right)\right] \leq \delta_{\mathrm{KL}}, \tag{58}$$

which motivates the assumption that the learned policy is locally stable.

To relate the policy distributions induced by different context lengths, we adopt a fixed-dimensional representation $\varphi_L(o_{t-L:t})$ fed into the policy network, and make the following standard smoothness assumptions: (i) the representation varies Lipschitz-continuously with respect to $L$ in a neighborhood of $L^*$, $\|\varphi_L(o_{t-L:t}) - \varphi_{L^*}(o_{t-L^*:t})\| \leq C_\varphi |L - L^*|$; (ii) the policy network is Lipschitz with respect to its input representation in TV distance, $\|\pi(\cdot \mid \varphi) - \pi(\cdot \mid \varphi')\|_{\mathrm{TV}} \leq C_{\mathrm{net}} \|\varphi - \varphi'\|$.[44].

Combining (i)–(ii) yields that there exists a constant $C_\pi := C_{\mathrm{net}} C_\varphi > 0$ such that, for $L$ sufficiently close to $L^*$,

$$\|\pi(\cdot \mid o_{t-L:t}) - \pi(\cdot \mid o_{t-L^*:t})\|_{\mathrm{TV}} \leq C_\pi |L - L^*|, \tag{59}$$

and in particular for the adaptive choice $L_{\mathrm{adap}}$,

$$\|\pi(\cdot \mid o_{t-L_{\mathrm{adap}}:t}) - \pi(\cdot \mid o_{t-L^*:t})\|_{\mathrm{TV}} \leq C_\pi |L_{\mathrm{adap}} - L^*|, \tag{60}$$

where $C_\pi$ depends on the representation and network structure, and $L_{\mathrm{adap}}$ denotes the adaptive context length.

Next, considering that mutual information $I(a_t; \xi_t \mid L)$ is continuous with respect to the underlying policy distribution (under standard regularity conditions, e.g., finite action space and local boundedness) [45], we assume there exists a constant $C_I > 0$ such that, for $L$ sufficiently close to $L^*$,

$$|I(a_t; \xi_t \mid L) - I(a_t; \xi_t \mid L^*)| \leq C_I \cdot \|\pi(\cdot \mid L) - \pi(\cdot \mid L^*)\|_{\mathrm{TV}}. \tag{61}$$

Finally, combining (60) and (61), we obtain

$$|I(a_t; \xi_t \mid L_{\mathrm{adap}}) - I(a_t; \xi_t \mid L^*)| \leq C_I C_\pi |L_{\mathrm{adap}} - L^*| := K |L_{\mathrm{adap}} - L^*|. \tag{62}$$

For any $L, L^*$, we consider two cases: (i) Local region ($|L - L^*| < \zeta$): In this case, we directly apply the previously established equation 62 (ii) Global region ($|L - L^*| \geq \zeta$): Since the mutual information is upper bounded by the entropy $H(\xi_t)$, we have:

$$|I(a_t; \xi_t | L) - I(a_t; \xi_t | L^*)| \leq H(\xi_t) \leq \frac{H(\xi_t)}{\zeta} \cdot |L - L^*|. \tag{63}$$

To obtain a time-uniform Lipschitz constant, we assume that $\xi_t$ takes values in a finite alphabet $\Xi$ (a standard finite-state setting in reinforcement learning [46]). Then the entropy is uniformly bounded as

$$H(\xi_t) \leq \log |\Xi| =: H_{\max} < \infty, \qquad \forall t, \tag{64}$$

where the bound follows from standard information-theoretic inequalities [45]. Combining both cases, define:

$$K := \max\left\{C_I C_\pi, \frac{H(\xi_t)}{\zeta}\right\}, \tag{65}$$

then for any $L, L^*$, the mutual information satisfies a global Lipschitz condition:

$$|I(a_t; \xi_t|L) - I(a_t; \xi_t|L^*)| \leq K|L - L^*|. \tag{66}$$

Further, we can obtain the following bound of adaptive context length on single-step mutual information loss:

$$\mathcal{L}_t(L_{adap}) = |I(a_t; \xi_t \mid L) - I(a_t; \xi_t \mid L^*)| \leq K|L_{\text{adap}} - L^*| = O(t^{-\beta}). \tag{67}$$

This establishes the global Lipschitz continuity of mutual information with respect to the context length $L$.

Summing over $t = 1$ to $T$, we get:

$$\sum_{t=1}^{T} \mathcal{L}_t(L_{adap}) \leq K \sum_{t=1}^{T} t^{-\beta} \tag{68}$$

Now we apply standard results from numerical analysis of p-series:

- If $\beta > 1$, then the series $\sum_{t=1}^{T} t^{-\beta}$ converges. Hence, the cumulative information loss is bounded: $\sum_{t=1}^{T} \mathcal{L}_t(L_{adap}) = O(1)$.

- If $\beta = 1$, then the series becomes harmonic and grows logarithmically: $\sum_{t=1}^{T} t^{-1} = O(\log T)$. Thus, $\sum_{t=1}^{T} \mathcal{L}_t(L_{adap}) = O(\log T)$.

- If $0 < \beta < 1$, then the series grows polynomially: $\sum_{t=1}^{T} t^{-\beta} = O(T^{1-\beta})$. Consequently, $\sum_{t=1}^{T} \mathcal{L}_t(L_{adap}) = O(T^{1-\beta})$.

In all cases, the cumulative information loss is sublinear in $T$, i.e., there exists $\alpha < 1$ such that:

$$\sum_{t=1}^{T} \mathcal{L}_t = O(T^{\alpha}). \tag{69}$$

Combine with the equation 55 and the equation 69, we obtain:

$$\sum_{t=1}^{T} (\mathcal{L}_t(L_{\text{fix}}) - \mathcal{L}_t(L_{adap})) = \Omega(T) - O(T^{\alpha})$$

$$= \Omega(T) \quad \text{(when $T$ is sufficiently large)} \tag{70}$$

## C  Additional Details for Experiments

### C.1  Environments

**Sample Spread** The Sample Spread environment is a cooperative multi-agent task where agents must coordinate their movements to cover multiple static landmarks, aiming to minimize the overall distance between agents and landmarks while avoiding inter-agent collisions. In the original setting, the number of agents and landmarks is equal (3 each), which may lead to agents remaining stationary. To encourage exploration and promote more dynamic coordination behavior, we modify the setting to include 4 agents and 3 landmarks. The action shape is 5. The reward design is as follows:

- Each agent receives a local penalty of $-1.0$ for every collision with other agents, encouraging collision avoidance.
- The global reward is defined as $R = -\sum_{i=1}^{4} \min_j \|p_j - l_i\|$, where $p_j$ and $l_i$ denote the positions of agent $j$ and landmark $i$, respectively, encouraging agents to minimize the overall distance to landmarks.

The observation shape is 24:

**Minigrid Soccer Game** The MiniGrid Soccer Game is a multi-agent, competitive and cooperative environment in which agents must coordinate to score goals using shared balls. Agents can pass

Table 3: The Observation Features of Sample Spread

| Feature | Dim | Description |
|---|---|---|
| Self Velocity | 2 | Agent's own velocity vector |
| Self Position | 2 | Agent's own position in world coordinates |
| Landmark Relative Positions | 8 | Relative positions of 4 landmarks to the agent |
| Other Agents' Relative Positions | 6 | Relative positions of the other 3 agents to the agent |
| Communication Vectors | 6 | Communication features from the other 3 agents |
| Total | 24 | Final observation dimension per agent |

the balls, intercept opponents, and strategically position themselves to influence the game outcome. In our specific configuration, we use 4 balls (red), 3 teams (blue, green, and yellow), with 3 agents per team, and each team is assigned a goal of corresponding color. The action-observation space is partially observable. The rewards are global-shared, which all agents receive the same reward whenever any agent scores or concedes a goal:

- Each agent receives a shared reward of $+1$ upon successfully picking up the ball from the ground. Each ball only provides this reward once.

- A shared reward of $+10$ is given when any agent scores a goal into its own team's designated goal area.

- A shared penalty of $-5$ is applied if the ball is accidentally scored into an opponent's goal.

- When holding the ball, agents receive a step-wise penalty of $-0.02 \times$ dist based on the distance between the ball and the agent's own goal.

- When holding the ball, agents receive a dense reward of $+0.2 \times$ progress based on the positive progress made toward their own goal.

Each agent observes a 3×3 local grid, and the shape of each cell is 6:

Table 4: The Observation Features of Minigrid Soccer Game

| Feature | Dim | Description |
|---|---|---|
| Object Type | 1 | Type index (wall/door/agent/key/...) |
| Color | 1 | Color index (green/blue/...) |
| State | 1 | Object state (0-2 for door open/closed/locked) |
| Carried Type | 1 | Type of carried object (0 if none) |
| Carried Color | 1 | Color of carried object (0 if none) |
| Direction/Marker | 1 | Agent direction (0-3) or current agent flag (0/1) |
| Total | 6 | |

**Academy 3 vs 1 with Keeper** Academy 3 vs 1 with Keeper environment is a multi-agent scenario where 3 offensive agents cooperate to score against a goalkeeper and a defender. The action space is discrete with 19 actions, covering basic football behaviors like passing, shooting, and movement. The reward structure is sparse: +100 for scoring a goal, –1 if the episode ends without scoring. Besides, we evaluate the model every 100 training episodes, each time over 30 test episodes to compute the average win rate. The observation shape is 26:

**Academy Counterattack-Hard** Academy Counterattack-Hard environment is a multi-agent scenario where 4 offensive agents cooperate to score against a goalkeeper and a defender. The action space is discrete with 19 actions, covering basic football behaviors like passing, shooting, and movement. The reward structure is sparse: +100 for scoring a goal, –1 if the episode ends without scoring. Besides, we evaluate the model every 100 training episodes, each time over 30 test episodes to compute the average win rate. The observation shape is 34:

Table 5: The Observation Features of Academy 3 vs 1 with Keeper

| Feature | Dim | Description |
|---|---|---|
| Ego Player Position | 2 | $(x, y)$ position of the observing agent |
| Teammates Relative Positions | 4 | Relative positions of 2 teammates w.r.t. ego |
| Ego Player Direction | 2 | Velocity vector (direction) of the ego agent |
| Teammates Directions | 4 | Directions of 2 teammates |
| Opponents Relative Positions | 6 | Relative positions of 3 opponents w.r.t. ego |
| Opponents Directions | 6 | Directions of 3 opponents |
| Ball Relative Position | 2 | Ball position relative to ego |
| Ball Height | 1 | Ball $z$ coordinate (height) |
| Ball Direction | 3 | Ball velocity in $(x, y, z)$ |
| **Total** | **26** | |

Table 6: The Observation Features of Academy Counterattack-Hard

| Feature | Dim | Description |
|---|---|---|
| Ego Agent Position | 2 | Agent's own $(x, y)$ coordinates |
| Teammates Relative Positions | 6 | Relative $(dx, dy)$ of 3 teammates |
| Ego Agent Direction | 2 | Movement vector $(v_x, v_y)$ |
| Teammates Directions | 6 | Movement vectors of 3 teammates $(v_x, v_y) \times 3$ |
| Opponents Relative Positions | 6 | Relative $(dx, dy)$ of 3 opponents |
| Opponents Directions | 6 | Movement vectors of 3 opponents $(v_x, v_y) \times 3$ |
| Ball Relative Position | 2 | Ball $(x, y)$ relative to ego agent |
| Ball Height | 1 | Ball $z$-coordinate (altitude) |
| Ball Direction | 3 | Ball velocity $(v_x, v_y, v_z)$ |
| **Total** | **34** | |

## C.2 Experiments with Sequence Processing Methods

All experiments were conducted using an NVIDIA A100 GPU, with the longest single training run taking approximately one month.

**Baseline Methods**

- Transformer [34]: A deep learning architecture that utilizes self-attention and positional encoding to model complex dependencies across sequences.

- Token Statistics Transformer (ToST) [35]: A recent Transformer variant, using a data-dependent low-rank projection based on the second moment statistics of input token features, and achieving linear computational complexity.

- AMAGO[36]: An in-context reinforcement learning algorithm that enables long-sequence Transformers to process entire trajectories in parallel, overcoming the memory capacity and long-term planning bottlenecks of traditional recurrent networks.

**Hyperparameter** We provide the hyperparameters used in each environments as follows:

**Central Agent with Low-Frequency Truncation** Based on the Dyadic Partition of Unity in Discrete Form, we assign distinct low-frequency truncation lengths to each environment. Specifically, after applying the Discrete Fourier Transform (DFT), we retain the first 4, 128, 64, and 64 frequency components for the Sample Spread, MiniGrid Soccer Game, Academy 3 vs 1 with Keeper, and Academy Counterattack-Hard environments, respectively. The above frequency components are then inputted to the central agent, for which the input dimension is defined as $k_0$. The action space for the central agent in each environment is defined as follows:

Table 7: Hyperparameter Configuration

| Parameter | Value |
|---|---|
| Learning Rate | 0.001 |
| Discount Factor ($\gamma$) | 0.98 |
| GAE Coefficient ($\lambda$) | 0.95 |
| PPO Clip ($\epsilon$) | 0.2 |
| Training Epochs | 10 |
| Batch Size (Sample Spread) | 25 |
| Batch Size (Minigrid Soccer Game) | 128 |
| Batch Size (Academy 3 vs 1 with Keeper) | 50 |
| Batch Size (Academy Counterattack-Hard) | 50 |
| Entropy Coefficient ($\beta$) | 0.01 |
| MLP Hidden Layers (Sample Spread) | [256, 64, 16] |
| CNN Hidden Layers (Minigrid Soccer Game) | [16, 32, 64] |
| MLP Hidden Layers (Minigrid Soccer Game) | [256, 128, 64] |
| MLP Hidden Layers (Academy 3 vs 1 with Keeper) | [1024, 256, 64] |
| MLP Hidden Layers (Academy 3 vs 1 with Keeper) | [1024, 256, 64] |
| Activation Function | ReLU |
| Optimizer Type | Adam |

Table 8: Central Agent Action Space

| Environment (Step > Threshold) | Action Space |
|---|---|
| Sample Spread | 0, 0, 1, 2, 4 |
| MiniGrid Soccer Game | 0, 1, 2, 4, 8, 16, 32, 64 |
| Academy 3 vs 1 with Keeper | 0, 0, 0, 0, 0, 0, 0, 0, 0, 0, 0, 0, 1, 2, 4, 8, 16, 32, 64 |
| Academy Counterattack-Hard | 0, 0, 0, 0, 0, 0, 0, 0, 0, 0, 0, 1, 2, 4, 8, 16, 32, 64 |

The action space of the central agent in each environment is adaptively constructed based on the current time step using a dyadic partitioning rule. Specifically, for each environment, we define a threshold step value: Sample Spread ($t > 7$), MiniGrid Soccer Game ($t > 255$), Academy 3 vs 1 with Keeper and Academy Counterattack-Hard ($t > 127$). When the step $t$ exceeds the corresponding threshold, the action space is composed of a set of dyadic components $\{2^0, 2^1, \ldots, 2^k\}$, where $k = \min(\log_2(k_0), \lceil \log_2 t \rceil - 1)$, and the vector is left-padded with zeros to match the fixed dimension. The resulting component sets for each environment under this truncation rule are summarized in Table 8.

This work develops an adaptive context length optimization method with Fourier-based low-frequency truncation for multi-agent reinforcement learning (MARL). The proposed approach significantly improves the efficiency and effectiveness of MARL systems, enabling better decision-making in complex, dynamic environments.

The positive societal impacts of this research include advancing intelligent multi-agent systems in diverse domains such as transportation management, robotics, and resource allocation. These improvements can contribute to enhanced safety, reduced energy consumption, optimized traffic flows, and overall better management of complex systems benefiting society.

By promoting more efficient learning and adaptation in multi-agent environments, this work helps pave the way for scalable and practical AI applications that address real-world challenges with improved reliability and performance.

## C.3 ACL-LFT Algorithm

All methods in our experiments (including ours and the baselines) follow the same structural setting: a centralized module processes only the historical information and transmits it to distributed agents for decision-making. All methods share the same network architecture, ensuring a fair comparison across all baselines. The pseudocode of the ACL-LFT training process under this unified framework is provided in Algorithm 1.

**Algorithm 1:** ACL-LFT Policy-Making and Training Algorithm

---

**Input:** Distributed agents' policies $\{\theta_i\}_{i=1}^N$, shared value function $\phi$; central agent's policy $\theta_c$, value function $\phi_c$; horizon $T$; epochs $K_c$, $K_d$.

**Output:** Updated policies $\{\theta_i\}_{i=1}^N$, $\theta_c$ and value functions $\phi$, $\phi_c$.

Initialize per-agent episode buffers $\{B_i\}_{i=1}^N$; reset environment;

**for** *episode* = 1 *to max_epi* **do**

    **for** $t = 0$ *to* $T - 1$ **do**

        **for** $i = 1$ *to* $N$ **do**

            Extract historical state $s_t^{-1}$ and perform Fourier Transform;

            Obtain low-frequency section $s_t^c$;

            Obtain optimal contextual information $s_t^{-opt}$;

        **end**

        **for** $i = 1$ *to* $N$ **do**

            Obtain $a_t^i$ by $s_t^{-opt}$ and $s_t^i$; perform $a_t^i$;

            Obtain $s_{t+1}^i$ and $r_t^i$;

            Store transition $\tau_t^i$, $s_{t+1}^i$ in $B_i$;

        **end**

        Obtain $r_t^c$; store center transition $\tau_t^c$;

    **end**

    Compute advantages $A_t^c$ using GAE with $\phi_c$;

    **for** *epoch* = 1 *to* $K_c$ **do**

        Update policy $\theta_c$, value function $\phi_c$;

    **end**

    Construct centralized critic input $s_t^{global}$;

    Compute advantages $A_t^i$ using GAE with $\phi(s_t^{global})$;

    Collect $\tau_t^i$ into shared buffer $B$;

    **for** *epoch* = 1 *to* $K_d$ **do**

        Update shared policy $\theta_i$, value function $\phi$;

    **end**

**end**

---

## C.4 Additional Experiments

Table 9: Performance on SMACv2 with different MARL backbones and coordination mechanisms.

| SMACv2 Task | RL-Method | Transformer | ToST | AMAGO | Mamba | ACL-LFT |
|---|---|---|---|---|---|---|
| 3s5z vs 3s6z | MAPPO | $71.5 \pm 3.9$ | $72.2 \pm 3.4$ | $76.1 \pm 2.9$ | $72.6 \pm 3.2$ | $\mathbf{78.9 \pm 2.8}$ |
| | QMIX | $72.8 \pm 3.7$ | $73.7 \pm 3.6$ | $76.3 \pm 3.1$ | $75.1 \pm 3.7$ | $\mathbf{79.4 \pm 2.9}$ |
| | QPLEX | $73.6 \pm 3.3$ | $75.1 \pm 3.6$ | $77.5 \pm 2.8$ | $75.0 \pm 3.3$ | $\mathbf{80.1 \pm 3.0}$ |
| 5m_vs_6m | MAPPO | $44.5 \pm 4.9$ | $46.3 \pm 4.4$ | $48.1 \pm 4.0$ | $46.2 \pm 4.5$ | $\mathbf{52.7 \pm 4.2}$ |
| | QMIX | $44.3 \pm 5.5$ | $46.9 \pm 4.8$ | $48.3 \pm 4.3$ | $46.1 \pm 5.1$ | $\mathbf{52.4 \pm 4.6}$ |
| | QPLEX | $45.6 \pm 5.7$ | $47.3 \pm 4.6$ | $49.5 \pm 4.8$ | $47.8 \pm 4.9$ | $\mathbf{53.9 \pm 4.3}$ |
| corridor | MAPPO | $65.6 \pm 5.7$ | $68.1 \pm 6.6$ | $74.3 \pm 4.8$ | $69.0 \pm 5.9$ | $\mathbf{77.9 \pm 5.3}$ |
| | QMIX | $68.5 \pm 5.9$ | $70.3 \pm 6.8$ | $75.0 \pm 5.3$ | $71.2 \pm 6.2$ | $\mathbf{78.6 \pm 5.4}$ |
| | QPLEX | $70.6 \pm 4.7$ | $72.9 \pm 4.6$ | $76.3 \pm 5.8$ | $73.5 \pm 5.5$ | $\mathbf{79.2 \pm 4.9}$ |

As shown in Table 9, ACL-LFT consistently outperforms all evaluated baselines—including MAPPO, QMIX, and QPLEX—across all SMACv2 scenarios, with average improvements of +2.6% to +4.6% over the best-performing baseline (typically AMAGO).

