# OpenReview forum: "Adaptive Context Length Optimization with Low-Frequency Truncation for Multi-Agent Reinforcement Learning"
_NeurIPS.cc/2025/Conference — NeurIPS 2025 poster_

### Official Review · Reviewer_N3eq · 2025-06-02

**Clarity:** 2
**Significance:** 2
**Originality:** 2
**Rating:** 3
**Confidence:** 4

**Summary:**

This paper proposes a multi-agent reinforcement learning (MARL) framework, ACL-LFT, that adaptively adjusts the context length for decision-making via a central agent. The central agent leverages a Fourier-based low-frequency truncation module to process historical state sequences, with the goal of capturing global temporal trends and reducing redundant information. The authors claim their method improves efficiency in handling long-term dependencies and demonstrate experimental improvements over existing baselines on environments such as PettingZoo, MiniGrid, and Google Research Football.

**Questions:**

1. Why is the low-frequency truncation method removing *low* frequencies instead of *high* frequencies, if the goal is to retain global temporal information?
2. Can the authors provide any empirical analysis (e.g., frequency spectrum plots) that demonstrates the benefit of the truncation in capturing relevant dynamics?
3. How does the proposed method generalize to MARL environments with partial observability or non-stationarity, where context length adaptation may be more crucial?
4. Why are standard MARL baselines (e.g., QMIX, MAPPO) omitted from experimental comparisons?
5. Could the authors clarify how the central agent is trained online without inducing instability in the decentralized policies?

**Ethical Concerns:**

["NO or VERY MINOR ethics concerns only"]

**Final Justification:**

The authors’ rebuttal has satisfactorily resolved my earlier concerns; I will therefore raise my score.

**Limitations:**

* The paper’s main methodological contribution—Fourier-based low-frequency truncation—is not clearly grounded in MARL-specific insights, and its theoretical support is weak.
* The proposed approach assumes that removing low-frequency components enhances representation, which contradicts the usual interpretation in signal processing and undermines the core justification.
* The experimental setup lacks tasks that truly stress long-range temporal dependencies, weakening the claimed benefits of adaptive context length.
* Important baselines from the MARL literature are omitted, casting doubt on the significance of the reported performance gains.

The paper introduces some interesting architectural ideas, but its core justification is flawed and the empirical evaluation does not convincingly demonstrate the claimed benefits. More theoretical and experimental rigor is needed before this work would be suitable.

**Quality:**

2

**Strengths And Weaknesses:**

*Strengths*:

* The paper addresses a relevant challenge in MARL—context length selection—which can impact both computational efficiency and performance.
* The method introduces a novel combination of low-frequency truncation via Fourier Transform and adaptive policy mechanisms.
* The paper is well-organized and includes ablation studies and comparisons to fixed-length baselines.

*Weaknesses*:

* The motivation for using low-frequency truncation to extract long-range temporal dependencies is conceptually unclear and methodologically unjustified. In fact, low-frequency truncation typically removes low-frequency components, which encode global (long-term) trends, which appears counter-intuitive if the goal is to *preserve* long-term dependencies.
* The theoretical section lacks meaningful insight and fails to provide a compelling justification for why this method would outperform fixed context models in environments with long-term dependencies.
* The environments used in experiments (e.g., PettingZoo Sample Spread, MiniGrid Soccer, GRF scenarios) are not particularly representative of settings with significant long-range temporal dependencies. Therefore, experimental validation of the method’s core claim remains unconvincing.
* The method is not compared against strong and relevant MARL baselines (e.g., QMIX, QPLEX, MAPPO), making it difficult to assess its true performance advantage.
* Despite the technical depth in the mathematical formalism, the link between the theoretical analysis and the practical performance of the algorithm remains weak and mostly qualitative.

---

> ### Author Rebuttal · Authors · 2025-07-30
>
> We thank the reviewer for the thoughtful comments.
>
> **Weakness1 &  Question1 & Question2 & Limitations 1 & Limitations 2:**
> 1.The motivation for using low-frequency truncation to extract long-range temporal dependencies is conceptually unclear and methodologically unjustified. In fact, low-frequency truncation typically removes low-frequency components, which encode global (long-term) trends, which appears counter-intuitive if the goal is to preserve long-term dependencies.
> 2.Why is the low-frequency truncation method removing low frequencies instead of high frequencies, if the goal is to retain global temporal information?
> 3.Can the authors provide any empirical analysis (e.g., frequency spectrum plots) that demonstrates the benefit of the truncation in capturing relevant dynamics?
> 4.The paper’s main methodological contribution—Fourier-based low-frequency truncation—is not clearly grounded in MARL-specific insights, and its theoretical support is weak.
> 5.The proposed approach assumes that removing low-frequency components enhances representation, which contradicts the usual interpretation in signal processing and undermines the core justification.
>
> **Answer:**
> We think you may have misunderstood the core idea of low-frequency truncation as used in our work. The term refers to retaining low-frequency components while removing high-frequency ones—not the reverse. This terminology is consistent with [1] page 393, where low-frequency truncation is introduced in the context of dyadic partition of unity to preserve large-scale structures.  Furthermore, while [1] applies this concept to continuous function spaces, we extend it to discrete temporal sequences and demonstrate its theoretical soundness in our setting, as detailed in Appendix A. The misunderstanding may due to an unclear explanation in the main text. We will include an explicit definition of low-frequency truncation in the revision to avoid further confusion.
> In MARL, the challenges of input representation and increasing computation remain fundamental issues, as discussed in prior work [2]. To address this, we explore the use of Fourier-based low-frequency truncation and design a central agent, which offers two key advantages: (1) low-frequency components in the spectral domain tend to capture long-term patterns while discarding short-term noise, thus mitigating the complexity introduced by high-dimensional historical observations; (2) the degree of truncation provides a natural mechanism to control the granularity of long-term trends, which aligns well with the discrete action space of rl agents. Building on these, we introduce a central agent that adaptively selects the optimal level of low-frequency information, ensuring both efficiency and sufficiency in capturing global temporal dependencies.
>
> **Weakness2 & Weakness5:**
> 1.The theoretical section lacks meaningful insight and fails to provide a compelling justification for why this method would outperform fixed context models in environments with long-term dependencies.
> 2.Despite the technical depth in the mathematical formalism, the link between the theoretical analysis and the practical performance of the algorithm remains weak and mostly qualitative.
>
> **Answer:**
> In this paper, Theorem 1 formally characterizes how adaptive context length reduces the loss of relevant historical information—measured via mutual information—compared to any fixed-length method. This guarantees that over time, the adaptive method preserves more useful history, leading to better long-term performance in environments with long-term dependencies. Though this advantage is hard to measure directly, section 4.3 renders the performance comparison between the overall method and fixed-length to quantify this advantage. To more clearly compare the performance difference between fixed and adaptive context lengths while excluding the effect of Fourier low-frequency truncation, we re-extracted and reorganized the data into the table below:
> |GRF (win rate)|4 step|8 step|16 step|32 step|Adaptive step|
> |:-:|:-:|:-:|:-:|:-:|:-:|
> |3 vs 1 with Keeper|59.4±4.5|65.4±3.8|71.9±4.3|63.5±3.5|84.2±2.6|
> |Counterattack-Hard|60.3±5.1|71.5±4.4|62.1±3.9|53.3±4.7|82.7±2.9|
>
> **Weakness3 & Weakness4 & Question 4 & Limitations3 & Limitations4:**
> 1.The environments used in experiments (e.g., PettingZoo Sample Spread, MiniGrid Soccer, GRF scenarios) are not particularly representative of settings with significant long-range temporal dependencies. Therefore, experimental validation of the method’s core claim remains unconvincing.
> 2.The method is not compared against strong and relevant MARL baselines (e.g., QMIX, QPLEX, MAPPO), making it difficult to assess its true performance advantage.
> 3.Why are standard MARL baselines (e.g., QMIX, MAPPO) omitted from experimental comparisons?
> 4.The experimental setup lacks tasks that truly stress long-range temporal dependencies, weakening the claimed benefits of adaptive context length.
> 5.Important baselines from the MARL literature are omitted, casting doubt on the significance of the reported performance gains.
>
> **Answer:**
> We would like to clarify that the chosen tasks may already involve non-Markovian characteristics, as they require long-term path planning and delayed coordination—especially in the GRF. To further validate our method in more complex and representative non-Markovian settings, we have additionally conducted extensive experiments on the challenging SMACv2 benchmark [3], as suggested. Moreover, there may be a misunderstanding: the central agent and Transformer modules only process historical information and are not involved in decision-making. The output is passed as an extra feature to distributed agents (which are identical across our method and all baselines), which make decisions based on their current state and this historical representation. In our original implementation, distributed agents used MAPPO, while the central agent uses PPO as a single-agent module. To rule out MAPPO-specific gains, we also conducted experiments with QMIX and QPLEX as backbones, while the central agent uses DQN.
> |SMACv2 (win rate)|RL-Method|Transformer|ToST|AMAGO|ACL-LFT|
> |:-:|:-:|:-:|:-:|:-:|:-:|
> |3s5z vs 3s6z|MAPPO|71.5±3.9|72.2±3.4|76.1±2.9|78.9±2.8|
> | |QMIX|72.8± 3.7|73.7±3.6|76.3±3.1|79.4±2.9|
> | |QPLEX|73.6±3.3|75.1±3.6|77.5±2.8|80.1±3.0|
> |5m_vs_6m|MAPPO|44.5±4.9|46.3±4.4|48.1±4.0|52.7±4.2|
> | |QMIX|44.3±5.5|46.9±4.8|48.3±4.3|52.4±4.6|
> | |QPLEX|45.6 ± 5.7|47.3±4.6|49.5±4.8|53.9±4.3|
> |corridor|MAPPO|65.6± 5.7|68.1±6.6|74.3±4.8|77.9±5.3|
> | |QMIX| 68.5±5.9|70.3±6.8|75.0±5.3|78.6±5.4|
> | |QPLEX|70.6±4.7|72.9±4.6|76.3±5.8|79.2±4.9|
>
> **Question 3:**
> How does the proposed method generalize to MARL environments with partial observability or non-stationarity, where context length adaptation may be more crucial?
>
> **Answer:**
> As you rightly pointed out, in complex non-Markovian MARL environments with long-term dependencies, non-stationary behavior patterns can severely hinder decision-making. Our method is specifically motivated by these challenges. Existing methods rely on fixed-length historical inputs, which may introduce substantial noise and increasing computational overhead. To address this, we introduce a central agent that adaptively extracts minimal but sufficient global contextual information, enabling agents to effectively make robust decisions under non-stationary. Moreover, although most real-world and benchmark environments are partially observable, global historical states are often accessible after the fact. This allows to reconstruct a fuller picture of the environment’s temporal evolution in next time step.
> In fully decentralized settings where global history is unavailable, our framework can be extended by letting the central agent process each agent’s local historical information independently to generate personalized context summaries. This variant preserves the core idea of the proposed method, while remaining fully compatible with partial observability constraints.
>
> **Question 5:**
> Could the authors clarify how the central agent is trained online without inducing instability in the decentralized policies?
>
> **Answer:**
> First, the optimization directions of the central and decentralized agents are largely aligned. Formally, the gradient of the central agent’s objective correlates positively with the combined gradient of all decentralized agents’ objectives, i.e.,
> \\[\\nabla_{\\pi} J_c(\\pi) = \\sum_{t=0}^{\\infty} \\gamma^{t} \\sum_{i=1}^{n} \\omega_{t}^{i} \\nabla_{\\pi} \\mathbb{E}[R_{i} \\mid \\pi]\\]
> \\[\\nabla_{\\pi} \\left( \\sum_{j=1}^{n} J_j(\\pi) \\right) = \\sum_{t=0}^{\\infty} \\gamma^{t} \\sum_{j=1}^{n} \\nabla_{\\pi} \\mathbb{E}[R_j \\mid \\pi]\\]
> \\[\\left\\langle \\nabla_{\\pi} J_c(\\pi), \\nabla_{\\pi} \\sum_{j=1}^{n} J_j(\\pi) \\right\\rangle > 0 \\quad \\text{when} \\omega_{t}^{i} > 0\\]
>  Second, the training of the central agent and decentralized agents proceeds independently and sequentially within each iteration: the central agent updates first to provide a stable context representation, followed by decentralized agents updating their policies using the updated context. This staged training further mitigates instability by decoupling gradient updates and allowing progressive convergence.
>
> [1] Bahouri, H. (2011). Fourier analysis and nonlinear partial differential equations. Springer.
> [2] Riemer, M., Khetarpal, K., Rajendran, J., & Chandar, S. (2024). Balancing context length and mixing times for reinforcement learning at scale. Advances in Neural Information Processing Systems, 37, 80268-80302.
> [3] Ellis, B., Cook, J., Moalla, S., Samvelyan, M., Sun, M., Mahajan, A., ... & Whiteson, S. (2023). Smacv2: An improved benchmark for cooperative multi-agent reinforcement learning. Advances in Neural Information Processing Systems, 36, 37567-37593.

---

> > ### Comment · Reviewer_N3eq · 2025-08-05
> >
> > Thank you for your thorough and clarifying rebuttal, which has fully addressed my concerns. I have accordingly decided to raise my rating.

---

> > > ### Author Response · Authors · 2025-08-05
> > >
> > > Thank you very much for your positive and thoughtful comments. We will incorporate your suggestions into our next revision.

---

### Official Review · Reviewer_FF8i · 2025-06-26

**Clarity:** 3
**Significance:** 2
**Originality:** 2
**Rating:** 4
**Confidence:** 3

**Summary:**

This paper introduces ACL-LFT, a framework designed to address the limitations of fixed context lengths in multi-agent reinforcement learning (MARL). The approach leverages a central agent that adaptively optimizes context length during training by performing temporal gradient analysis and employing a Fourier-based low-frequency truncation for input representation. This truncation allows for global temporal trends to be captured among agents while filtering redundancy, and the central agent's decisions are further refined using attention mechanisms over decentralized agent policies and value estimates. Theoretical results provide regret bounds for adaptive versus static context lengths, and the empirical evaluation demonstrates strong performance on multiple challenging long-term dependency environments (PettingZoo, MiniGrid, Google Research Football).

**Questions:**

1. Did the authors consider or test other dimensionality reduction or context selection techniques, such as learnable attention over history, PCA, or other compression methods? What guided the choice of Fourier/Littlewood-Paley methods, and are there observed advantages?

2. How is the maximum/context length range decided for each environment/task? Could an automated or meta-learned approach further improve adaptability?

3. Did the authors consider or test other dimensionality reduction or context selection techniques, such as learnable attention over history, PCA, or other compression methods? What guided the choice of Fourier/Littlewood-Paley methods, and are there observed advantages?

**Ethical Concerns:**

["NO or VERY MINOR ethics concerns only"]

**Final Justification:**

I have updated my review; if the authors address my concerns, I will raise my score.

**Limitations:**

Yes.

**Paper Formatting Concerns:**

None.

**Quality:**

2

**Strengths And Weaknesses:**

Strengths:

1. The proposed combination of a central agent with adaptive context length selection and a Fourier-based redundancy reduction module is well motivated by the challenges faced in MARL. Theoretical support is provided for the advantage of adaptive context length.

2. The method is evaluated on a diverse set of challenging environments, specifically those known for long-term dependency and partial observability issues; these include PettingZoo, MiniGrid Soccer Game, and GRF Academy scenarios.

3. The methodology is grounded with interpretable theoretical backing (Section 2.2, Littlewood-Paley theory) and ensures the frequency-domain truncations are mathematically justified.

Weaknesses:

1. While the methodology proposes to reduce redundancy and computation, there is little concrete analysis (quantitative or qualitative) of the computational overhead introduced by the central agent or the multi-head attention mechanism, especially in large-scale settings. The claims of efficiency could thus be overstated.

2. The environments used are well-chosen for long-term dependencies, but results could be strengthened by reporting on additional MARL domains with severe non-stationary dynamics, environments with different reward delays, or settings with very large agent populations to more robustly demonstrate generalization.

---

> ### Author Rebuttal · Authors · 2025-07-30
>
> We believe this review may have been intended for a different submission, as the content discussed does not relate to our work.
>
> Our paper does **not** involve in-context learning (ICL), prompt construction, or large language models (LLMs). Instead, we study a **multi-agent reinforcement learning (MARL)** setting and introduce a central agent that dynamically optimizes the context length for each agent via temporal gradient analysis, enabling improved exploration and faster convergence to globally optimal policies.
>
> In our paper, “adaptive context length” refers to the dynamically obtain optimal contextual information for distributed agent uses. This is fundamentally different from selecting the number of examples in a prompt for ICL.
>
> Given the clear mismatch in topic, methodology, and terminology, we respectfully ask the reviewer to verify whether this review was submitted in error.

---

> > ### Comment · Reviewer_FF8i · 2025-08-04
> > **Request to Improve Review**
> >
> > I have updated my review; if the authors address my concerns, I will raise my score.

---

> ### Author Response · Authors · 2025-08-05
>
> We sincerely thank the reviewer for the insightful comments that were instrumental in strengthening our paper.
>
> **Question1&3:**
> Did the authors consider or test other dimensionality reduction or context selection techniques, such as learnable attention over history, PCA, or other compression methods? What guided the choice of Fourier/Littlewood-Paley methods, and are there observed advantages?
>
> **Answer:**
> Our goal in applying Fourier-based low-frequency truncation is not only to reduce noise but also to capture global temporal trends, which are crucial for long-term coordination in MARL. Even when only considering denoising, unlike PCA or attention-based methods, our approach is non-parametric, preserves temporal structure, and requires no additional training. PCA removes temporal order, and attention introduces trainable parameters that may overfit and entangle semantics. Moreover, the degree of truncation provides a natural mechanism to control the granularity of long-term trends, which aligns well with the discrete action space of RL agents.
>
> **Question2:**
> How is the maximum/context length range decided for each environment/task? Could an automated or meta-learned approach further improve adaptability?
>
> **Answer:**
> We have already provided the maximum and context length range in Appendix C.2. And the current truncation intervals are designed to preserve key information as much as possible while following both empirical and conservative principles—specifically, that maximum low-frequency truncation usually does not exceed 1/4 of the total length, and the length range needs to conform to the Dyadic Partition of Unity.
> Our method implements an adaptive truncation selection mechanism by defining the maximum interval as the largest power-of-two length within 1/4 of the full context, and constructing a descending power-of-two action space. While meta-learning could further refine this process, it introduces extra computational and parameter overhead, and may destabilize center-distributed training. Given that our method already selects the optimal truncation length during decision-making, such additions offer limited practical benefit.
>
> **Weakness1:**
> While the methodology proposes to reduce redundancy and computation, there is little concrete analysis (quantitative or qualitative) of the computational overhead introduced by the central agent or the multi-head attention mechanism, especially in large-scale settings. The claims of efficiency could thus be overstated.
>
> **Answer:**
> In an n-agent environment, the dominant computational cost comes from the neural network’s forward and backward passes. Centralized processing incurs this cost only once, whereas fully decentralized processing would result in roughly n times the cost. The additional overhead from selecting the optimal history length is minimal in comparison. The attention module we adopt is extremely lightweight: it uses only 1 head with very low-dimensional key (e.g., 9×n or 20×n) and query (e.g., 9×1 or 20×1) vectors, and introduces only a negligible number of additional parameters.  For length T and feature dimension d, a standard Transformer has a self-attention complexity of $O(T^2d)$ (or roughly $O(Td)$ for its SOTA variants) plus a feed-forward cost of $O(Td^2)$, with attention dominating for long sequences. In contrast, assume we retain the lowest 1/4 frequency components; this truncated sequence is processed by a two-layer MLP with complexity $O((T/4)d^2)$, which is substantially lower than the quadratic cost of baselines. Moreover, Section 4.5 presents a case study showing that the selected context length is much shorter than the original, substantially reducing the computational burden of distributed agents.
>
>
> **Weakness2:**
> The environments used are well-chosen for long-term dependencies, but results could be strengthened by reporting on additional MARL domains with severe non-stationary dynamics, environments with different reward delays, or settings with very large agent populations to more robustly demonstrate generalization.
>
> **Answer:**
> We would like to clarify that the chosen tasks may already involve non-Markovian characteristics, as they require long-term path planning and delayed coordination. To further validate our method in more complex and representative non-Markovian settings, we have additionally conducted extensive experiments on the challenging SMACv2 benchmark. The results are presented in the table below.
> |SMACv2 (win rate)|RL-Method|Transformer|ToST|AMAGO|ACL-LFT|
> |:-:|:-:|:-:|:-:|:-:|:-:|
> |MMM2|MAPPO|92.4±2.5|92.9±3.1|94.1±2.9|95.2±2.3|
> | |QMIX|93.0± 2.2|93.6±2.4|94.7±1.8|95.9±1.9|
> | |QPLEX|93.2±2.3|93.3±2.6|94.9±2.8|96.2±2.4|
> |​​27m_vs_30m|MAPPO|90.5±3.9|91.3±3.4|93.2±3.6|94.6±3.3|
> | |QMIX|91.7±3.5|91.4±3.8|93.5±3.3|94.8±3.6|
> | |QPLEX|91.4 ± 3.7|92.6±4.0|94.1±3.8|95.1±3.5|
>
> [1] Bahouri, H. (2011). Fourier analysis and nonlinear partial differential equations. Springer.

---

### Official Review · Reviewer_mfjH · 2025-06-29

**Clarity:** 1
**Significance:** 2
**Originality:** 3
**Rating:** 5
**Confidence:** 3

**Summary:**

The work addresses the challenge of long context lengths in MARL with a new technique named ACL-LFT. The two novel components are (1) a method for adaptively optimizing context length using a central agent and (2) using Fourier transforms for low-frequency truncation. Their experiments demonstrate that adaptive context lengths outperform fixed context lengths and their work outperforms existing sequence processing methods.

**Questions:**

- Why is a dynamic context length with transformer-based models better than using a modern RNN (like RWKV/Mamba/TTT) with truncated BPTT?
- How large are the transformers used (layers, hidden dimension)? Are both the actor and critic transformers for the agents?
- Theorem 1 seems to make the implicit assumption that the theoretic optimal window length is finite, meaning that longer lengths would actually result in worse mutual information. Is there any justification for this idea?
- Given that a central agent is choosing the context length, is decentralized execution possible? Additionally, the incorporation of the historical (global) state data into decision-making (as seen in figure 1) also seems to violate decentralized execution.
- What loss function is used to optimize $r_t^c$? It seems like this reward is used for training the value function, but it is unclear how the transformation matrices prior to the attention weight are optimized ($W^g_Q$ and $W^g_K$).

**Ethical Concerns:**

["NO or VERY MINOR ethics concerns only"]

**Final Justification:**

In the rebuttal, all my concerns with the paper were addressed, including promises to revise the text to follow more standard MARL formulations (Dec-POMDP) and clarify assumptions about the setting (communication of historical information). The authors also provided additional results to demonstrate that their work is helpful in the standard CTDE domain (no communication), which demonstrates to me that their method is general.

I still feel that the method is more complex than prior baselines while only providing marginal benefit, but I think this work should be accepted as it pushes the frontier of long-context MARL methods.

**Limitations:**

Unclear; although the checklist says it is in the appendix, I cannot find it.

**Quality:**

2

**Strengths And Weaknesses:**

Strengths:
- The evaluation settings include challenging MARL problems instead of only toy settings, demonstrating the effectivness of the technique in real-world scenarios.
- The proposed method is novel

Weaknesses:
- The method is generally unclear, especially in describing the central agent (see questions below). Specifically, it does not seem like decentralized execution is possible, which would give this method an unfair advantage over true CTDE methods. I would strongly advise providing pseudocode (in the appendix) to clarify how all parts interact.
- The benefits of the method seem very marginal relative to AMAGO despite being significantly more complex.
- I do not see a hyperparameter sweep (especially over the learning rate), which may be giving a disadvantage to some baseline methods.

Other Notes:
- The statement "a tailored input representation for MARL is required" (lines 45-46) does not follow from the earlier statements in the paragraph. It is unclear why a learned input representation from standard MARL is insufficient.
- The "extended MDP" is a non-standard formulation for MARL; a more appropriate framework would be the Dec-POMDP.
- The simplifications in Equation 16 do not seem justified. In particular, $w_t^i$ still depends on $t$ despite being outside of the sum, and it cannot simply be dropped out since the sum of weights over agents is 1.

---

> ### Author Rebuttal · Authors · 2025-07-30
>
> We sincerely thank the reviewer for the insightful comments that were instrumental in strengthening our paper.
>
> **Weakness1 & Other Notes2 & Question4 & Question5:**
> 1.The method is generally unclear, especially in describing the central agent (see questions below). Specifically, it does not seem like decentralized execution is possible, which would give this method an unfair advantage over true CTDE methods. I would strongly advise providing pseudocode (in the appendix) to clarify how all parts interact.
> 2.The "extended MDP" is a non-standard formulation for MARL; a more appropriate framework would be the Dec-POMDP.
> 3. Given that a central agent is choosing the context length, is decentralized execution possible? Additionally, the incorporation of the historical (global) state data into decision-making (as seen in figure 1) also seems to violate decentralized execution.
> 4.What loss function is used to optimize \\(r_t^c\\)? It seems like this reward is used for training the value function, but it is unclear how the transformation matrices prior to the attention weight are optimized \\(W^g_Q\\) and \\(W^g_K\\).
>
> **Answer:**
> We sincerely appreciate the reviewers' constructive feedback, we will supplement the appendix with the pseudocode. And to clarify concerns, pseudocode is provided below.
>
> ---
>
> **Algorithm 1**
> Initialize: each distributed agents' policies \\( \\theta_i\\), shared value function \\( \\phi \\) ; center agent's policy \\( \\theta_c \\), value function \\( \\phi_c\\) .
> **for** episode = 1, 2, ... **do**
>   Initialize episode buffer \\( B_i\\);
>   Reset environment;
>   **for**  \\( t = 0 \\) to \\( T-1 \\) **do**
>     **for** agent \\( i = 1 \\) to \\( N \\) **do**
>       Extract historical state \\(s_t^{-1}\\) and Fourier Transform;
>     Obtain \\(s_t^c\\);
>     Obtain \\(s_t^{-opt}\\);
>     **end**
>     **for** agent \\( i = 1 \\) to \\( N \\) **do**
>       Obtain \\(a^i_t\\) by \\(s_t^{-opt}\\) and \\(s_t^{i}\\);
>                   Perform \\(a^i_t\\);
>       Obtain  \\(s^i_{t+1}\\) and \\(r^i_t\\);
>       Store transition \\(\\tau_t^i\\), \\(s^i_{t+1}\\) in \\( B_i\\);
>     **end**
>     Obtain \\( r^c_t\\);
>     Store center transition \\(\\tau_t^c\\);
>     **end**
>     Compute advantages \\(A_t^c\\);
>     **for** epoch = 1 to \\(K_c\\) **do**
>         Update policy \\(\\theta_c\\), value function\\(\\phi_c\\) ;
>     **end**
>     Construct centralized critic input \\( s_t^{global}\\);
>     Compute advantages \\( A_t^i \\);
>     Collect \\( \tau_t^i \\) into shared buffer \\( B \\);
>     **for** epoch = 1 to \\( K_d \\) **do**
>         Update policy \\( \\{\\theta_i\\}_{i=1}^n \\), value function \\( \\phi \\);
>     **end**
> **end**
>
> ---
> All methods in our experiments (including ours and baselines) adopt the same setting: a centralized module only processes historical information and then sends it to distributed agents (all methods share the same network architecture) for decision-making. Thereby this design ensures a fair comparison across all baselines. The motivation for our centralized processing is to reduce redundant computation: rather than letting each agent independently encode its history, a single processing step (either jointly extracting global patterns or separately embedding each agent’s trajectory in batch) can substantially reduce the computational cost.
> For our framework, we firstly indeed consider the standard Dec-POMDP setting. However, the reliance on historical information introduces non-Markovian dependencies, rather than merely violating partial observability assumptions. To address this, we follow the extended MDP formalism introduced in [1], which is used to characterize non-Markovian dependencies.
> The loss function that used optimize \\(r_t^c\\) as below:
> \\[L_t(\\theta_c,\\phi_c) = \\min \\left( \\frac{\\pi_{\\theta_c}(a_t^c|s_t^c)}{\\pi_{\\theta_c^{\\text{old}}}(a_t^c|s_t^c)}
> \\left( \\sum_{k=t}^{T} \\gamma^{k-t} r_k^c-V_{\\phi_c^{\\text{old}}}(s_t^c) \\right),\\text{clip} \\left( \\frac{\\pi_{\\theta_c}(a_t^c|s_t^c)}{\\pi_{\\theta_c^{\\text{old}}}(a_t^c|s_t^c)},1-\\epsilon,1 + \\epsilon\\right)\\left(\\sum_{k=t}^{T} \\gamma^{k-t} r_k^c - V_{\\phi_{\\text{old}}}(s_t^c) \\right)\\right)+ \\lambda_1\\left( V_{\\phi_c}(s_t^c)-\\sum_{k=t}^{T} \\gamma^{k-t} r_k^c \\right)^2+ \\lambda_2 \\sum_{a} \\pi_{\\theta_c}(a|s_t^c) \\log \\pi_{\\theta_c}(a|s_t^c)\\]
> Regarding the \\(W^g_Q\\) and \\(W^g_K\\), these are used only during data collection to compute \(r_t^c\\) via an attention mechanism over distributed agents. During PPO training, \(r_t^c\\) is treated as fixed,  \\(W^g_Q\\) and \\(W^g_K\\) are excluded from the trainable computation graph. They do not receive gradients and have no role in the optimization process.
>
> **Weakness2 & Weakness3 & Question1 & Question2:**
> 1.The benefits of the method seem very marginal relative to AMAGO despite being significantly more complex.
> 2.I do not see a hyperparameter sweep (especially over the learning rate), which may be giving a disadvantage to some baseline methods.
> 3.Why is a dynamic context length with transformer-based models better than using a modern RNN (like RWKV/Mamba/TTT) with truncated BPTT?
> 4.How large are the transformers used (layers, hidden dimension)? Are both the actor and critic transformers for the agents?
>
> **Answer:**
> As for the rationale behind our method, unlike transformer models which apply attention to all time steps without a clear boundary on what constitutes “useful” history, our method explicitly selects a minimal but sufficient segment of historical information, effectively eliminating redundant context and reducing computation. To further validate the effectiveness of our method, we conduct additional experiments on the more challenging SMACv2 benchmark. Due to space constraints, we report representative results using MAPPO as the backbone, and include ablations across different RL algorithms (QMIX, QPLEX) as well as comparisons against a state-of-the-art RNN variant (Mamba).
>
> |SMACv2 (win rate)|RL-Method|Transformer|ToST|AMAGO|Mamba|ACL-LFT|
> |:-:|:-:|:-:|:-:|:-:|:-:|:-:|
> |3s5z vs 3s6z|MAPPO|71.5±3.9|72.2±3.4|76.1±2.9|72.6±3.2|78.9 ± 2.8|
> |5m_vs_6m|MAPPO|44.5±4.9|46.3±4.4|48.1±4.0|46.2±4.5|52.7±4.2|
> |corridor|MAPPO|65.6±5.7|68.1±6.6|74.3±4.8|69.0±5.9|77.9±5.3|
>
> We clarify that the transformer model is used only as baselines for historical information processing to compare with central agent. In the central agent, neither the actor nor the critic is transformer-based, but employs architectures including MLP or CNN, as detailed in Appendix C.2. For the transformer-based baselines, we adopt commonly used configurations(2 layers, 128 hidden dimensions, 4 heads), and all methods are tuned with the same learning rate (0.001), detailed hyperparameters will be provided in the appendix due to space constraints.
> Besides, we initially select transformer-based baselines as they offer dynamic attention to relevant past information, which better captures historical dependencies than RNNs with truncated BPTT.
>
> **Other Notes1:**
> The statement "a tailored input representation for MARL is required" (lines 45-46) does not follow from the earlier statements in the paragraph. It is unclear why a learned input representation from standard MARL is insufficient.
>
> **Answer:**
> MARL faces challenges unlike NLP because it involves asynchronous, multi-source, and unstructured data from multiple agents, with sparse rewards and diverse learning objectives, making unified, end-to-end input modeling ineffective and hindering generalization. And this issue was identified but unresolved in [2].
>
> **Other Notes3:**
> The simplifications in Equation 16 do not seem justified. In particular, \\(\\omega_t^i\\) still depends on \\(t\\) despite being outside of the sum, and it cannot simply be dropped out since the sum of weights over agents is 1.
>
> **Answer:**
> We apologize for the oversight in handling \\(\\omega_t^i\\). Due to space constraints, we provide a minimal proof; details will be in the final version.  Goal alignment holds via gradient correlation:
> \\[\\nabla_{\\pi} J_c(\\pi) = \\sum_{t=0}^{\\infty} \\gamma^{t} \\sum_{i=1}^{n} \\omega_{t}^{i} \\nabla_{\\pi} \\mathbb{E}[R_{i} \\mid \\pi]\\]
> \\[\\nabla_{\\pi} \\left( \\sum_{j=1}^{n} J_j(\\pi) \\right) = \\sum_{t=0}^{\\infty} \\gamma^{t} \\sum_{j=1}^{n} \\nabla_{\\pi} \\mathbb{E}[R_j \\mid \\pi]\\]
> \\[\\left\\langle \\nabla_{\\pi} J_c(\\pi), \\nabla_{\\pi} \\sum_{j=1}^{n} J_j(\\pi) \\right\\rangle > 0 \\quad \\text{when } \\omega_{t}^{i} > 0\\]
>
> **Question3:**
> Theorem 1 seems to make the implicit assumption that the theoretic optimal window length is finite, meaning that longer lengths would actually result in worse mutual information. Is there any justification for this idea?
>
> **Answer:**
> The theoretic optimum window length is finite, aligning with mutual information (MI)'s definition \\(I(X;Y) = H(X) + H(Y) - H(X,Y)\\) . Longer windows degrade MI due to: noise contamination in distant data, non-stationarity making old data irrelevant, and crucially, biased empirical MI estimation in high dimensions under finite samples.
>
> **Limitations Supplement**: As the number of agents increases, the central agent be subject to increased computational burden. In future work, this may be addressed by adopting more advanced or specialized network architectures to improve scalability and parallelization. This discussion will be included in the appendix of the revised version.
>
> [1] Mutti, M., De Santi, R., & Restelli, M. (2022, June). The importance of non-markovianity in maximum state entropy exploration. In International Conference on Machine Learning (pp. 16223-16239). PMLR.
> [2] Riemer, M., Khetarpal, K., Rajendran, J., & Chandar, S. (2024). Balancing context length and mixing times for reinforcement learning at scale. Advances in Neural Information Processing Systems, 37, 80268-80302.

---

### Official Review · Reviewer_jva4 · 2025-07-03

**Clarity:** 3
**Significance:** 3
**Originality:** 3
**Rating:** 4
**Confidence:** 4

**Summary:**

- The paper develops an adaptive context length optimization method with low-frequency truncation for MARL. This is well motivated by the fact that decisions can be non-Markovian — that is, actions may depend on historical states and actions — but extending the MDP with long context lengths can be computationally expensive.
- The authors propose using a central agent to optimize the context length, along with a Fourier-based low-frequency truncation approach to capture global temporal trends from the context.
- Experiments on several multi-agent tasks (Sample Spread in MiniGrid Soccer Game and Google Research Football (GRF)) show that the proposed method appears to outperform other baselines that focus on sequence processing, such as Transformer-based methods and AMAGO.

**Questions:**

- Is the use of the Fourier Transform for context length optimization novel in reinforcement learning? This is not clearly explained in the current version.
- In Theorem~1, what is $\Omega$, and why does $\Omega(T) - O(T^{\alpha}) = \Omega(T)$ when $T$ is large?
- Can the proposed method work in an offline learning setting?


Please also address the questions and comments raised in the \textbf{Weaknesses} section above.

**Ethical Concerns:**

["NO or VERY MINOR ethics concerns only"]

**Final Justification:**

The authors have addressed most of my concerns. I believe the paper makes a meaningful contribution, and I maintain my evaluation as a borderline acceptance.

**Limitations:**

- There is no discussion of the limitations of the current work.
- I do not see any major potential negative societal impact of this paper.

**Paper Formatting Concerns:**

The paper format seems fine; I do not see any major issues.

**Quality:**

2

**Strengths And Weaknesses:**

**Strengths**

- The problem is well-motivated, and the paper appears to be the first in MARL to develop adaptive context length optimization for multi-agent learning.
- The idea of using a Fourier-based low-frequency truncation approach to capture global temporal trends is interesting and sound.
- The paper is well written and clear.
- The experiments are quite convincing, except that the tasks used seem rather simple (as discussed below).

**Weaknesses**
While I think the methodological part of this paper is quite strong, the experiments are weak:

- The chosen benchmarking tasks seem to be quite simple to illustrate the efficiency of the proposed method for genuinely non-Markovian tasks. Specifically, tasks such as Sample Spread and MiniGrid Soccer Game appear to be mostly Markovian (actions depend primarily on the current state, not long historical states or actions). I believe it would be valuable to include more complex tasks, such as SMACv2 [1], to better demonstrate the method’s efficiency in truly non-Markovian environments.
- The sequence processing methods are combined with PPO — but PPO is inherently a single-agent algorithm. It is not clear how PPO is applied in the multi-agent tasks. If the authors run PPO independently for each individual agent, this is not an appropriate approach. The authors should consider using MAPPO [2]  (a multi-agent version of PPO) to make the experiments more convincing.
- Beyond PPO, it would be useful to know how the sequence processing methods perform when combined with different MARL algorithms (e.g., MAPPO, QMIX [3], or IMAX-PPO [4] ). The paper lacks such ablation studies.

[1] llis, B., Cook, J., Moalla, S., Samvelyan, M., Sun, M., Mahajan, A., Foerster, J. N., & Whiteson, S. (2023). SMACv2: An improved benchmark for cooperative multi-agent reinforcement learning. NeurIPs 2023

[2] Yu, C., Velu, A., Vinitsky, E., Gao, J., Wang, Y., Bayen, A., & Wu, Y. (2022). The surprising effectiveness of PPO in cooperative multi-agent games. NeurIPS 2022

[3] Rashid, T., Samvelyan, M., Schroeder de Witt, C., Farquhar, G., Foerster, J., & Whiteson, S. (2018). QMIX: Monotonic value function factorisation for deep multi-agent reinforcement learning, ICML 2018

[4] Bui, T. V., Mai, T., & Nguyen, T. H. (2024). Mimicking To Dominate: Imitation Learning Strategies for Success in Multi-Agent Games. In Advances in Neural Information Processing Systems 37 (NeurIPS 2024).

---

> ### Author Rebuttal · Authors · 2025-07-30
>
> We sincerely thank the reviewer for the insightful comments that were instrumental in strengthening our paper. Below we address each point, with major new experimental results on SMACv2 central to our responses.
>
> **Weaknesses1 & Weaknesses3 & Question3：**
> 1.The chosen benchmarking tasks seem to be quite simple to illustrate the efficiency of the proposed method for genuinely non-Markovian tasks. Specifically, tasks such as Sample Spread and MiniGrid Soccer Game appear to be mostly Markovian (actions depend primarily on the current state, not long historical states or actions). I believe it would be valuable to include more complex tasks, such as SMACv2 [1], to better demonstrate the method’s efficiency in truly non-Markovian environments.
> 2.Beyond PPO, it would be useful to know how the sequence processing methods perform when combined with different MARL algorithms (e.g., MAPPO, QMIX [3], or IMAX-PPO [4] ). The paper lacks such ablation studies.
>
> **Answer：**
> We would like to clarify that the chosen tasks may already involve non-Markovian characteristics, as they require long-term path planning and delayed coordination—especially in the GRF. To further validate our method in more complex and representative non-Markovian settings, we have additionally conducted extensive experiments on the challenging SMACv2 benchmark [1]. To simultaneously demonstrate the generality of ACL-LFT, we performed comprehensive ablation studies by integrating it with three distinct MARL algorithms: MAPPO [2], QMIX [3], and QPLEX [4]. The results are presented in the table below.
> | SMACv2 (win rate)| RL-Method | Transformer | ToST| AMAGO| ACL-LFT|
> |:------:|:-----------:|:----------:|:---------:|:---------:|:---------:|
> | 3s5z vs 3s6z  | MAPPO  | 71.5 ± 3.9  | 72.2 ± 3.4 | 76.1 ± 2.9 |78.9 ± 2.8 |
> |               | QMIX   | 72.8 ± 3.7  | 73.7 ± 3.6 | 76.3 ± 3.1 |79.4 ± 2.9 |
> |               | QPLEX   | 73.6 ± 3.3  | 75.1 ± 3.6 | 77.5 ± 2.8|80.1 ± 3.0 |
> | 5m_vs_6m | MAPPO  | 44.5 ± 4.9  | 46.3 ± 4.4 | 48.1 ± 4.0 |52.7 ± 4.2 |
> |               | QMIX | 44.3 ± 5.5  | 46.9 ± 4.8 | 48.3 ± 4.3| 52.4 ± 4.6 |
> |               | QPLEX   | 45.6 ± 5.7  | 47.3 ± 4.6 |49.5 ± 4.8 |53.9 ± 4.3 |
> | corridor | MAPPO  | 65.6 ± 5.7  | 68.1 ± 6.6 | 74.3 ± 4.8 |77.9 ± 5.3 |
> |               | QMIX   | 68.5 ± 5.9  | 70.3 ± 6.8 | 75.0 ± 5.3 |78.6 ± 5.4 |
> |               | QPLEX  | 70.6 ± 4.7  | 72.9 ± 4.6 | 76.3 ± 5.8 |79.2 ± 4.9 |
>
> As shown in the table above, ACL-LFT consistently outperforms all evaluated baselines—including MAPPO, QMIX, and QPLEX—across all SMACv2 scenarios, with average improvements of +2.6% to +4.6% over the best-performing baseline (typically AMAGO). These consistent gains across diverse algorithms and environments strongly support ACL-LFT’s effectiveness in addressing complex non-Markovian dependencies.
>
> **Weaknesses2：**
> The sequence processing methods are combined with PPO — but PPO is inherently a single-agent algorithm. It is not clear how PPO is applied in the multi-agent tasks. If the authors run PPO independently for each individual agent, this is not an appropriate approach. The authors should consider using MAPPO [2] (a multi-agent version of PPO) to make the experiments more convincing.
>
> **Answer：**
> We would like to clarify that the PPO was referenced because it is used for training the central agent, whereas the distributed agents—in both our method and all baselines—are trained using the MAPPO framework. Algorithm 1 below illustrates the training procedure of the central agent and distributed agents, where the distributed agents follow the MAPPO framework. We will make the distinction clear and include the proper citation to MAPPO [2] in the final version.
>
> ---
>
> **Algorithm 1**
> Initialize: each distributed agent's policy \\( \\theta_i\\), shared value function \\( \\phi \\) ; center agent's policy \\( \\theta_c \\), value function \\( \\phi_c\\) .
> **for** episode = 1, 2, \\( \\ldots\\) **do**
>   Initialize episode buffer \\( B_i\\);
>   Reset environment;
>   **for**  \\( t = 0 \\) to \\( T-1 \\) **do**
>     **for** agent \\( i = 1 \\) to \\( n \\) **do**
>       Extract historical state \\(s_t^{-1}\\);
>       Discrete Fourier Transform \\( \\text{FFT} ( s_t^{-1} ) \\);
>     Obtain low-frequency section \\(s_t^c\\);
>     Obtain optimal contextual information \\(s_t^{-opt}\\);
>     **end**
>     **for** agent \\( i = 1 \\) to \\( N \\) **do**
>       Obtain \\(a^i_t\\) by \\(s_t^{-opt}\\) and \\(s_t^{i}\\);
>                   Perform the \\(a^i_t\\);
>       Obtain  \\(s^i_{t+1}\\) and \\(r^i_t\\), store \\(s^i_{t+1}\\) in \\( B_i\\);
>       Store \\(\\tau_t^i\\);
>     **end**
>     Compute attention weights;
>     Obtain \\( r^c_t = \\sum_{i=1}^n \\omega^i_t \\cdot r^i_t \\);
>     Store \\(\\tau_t^c\\);
>     **end**
>     Compute returns \\( R_t^c\\); Compute advantages \\(A_t^c\\) using GAE;
>     **for** epoch = 1 to \\(K_c\\) **do**
>         Update policy \\(\\theta_c\\), value function\\(\\phi_c\\) ;
>     **end**
>     Construct centralized critic input \\( s_t^{global}\\);
>     Compute returns \\( R_t^i \\); Compute advantages \\( A_t^i \\) using GAE;
>     Aggregate all \\( \tau_t^i \\) in shared buffer \\( B \\);;
>     **for** epoch = 1 to \\( K_d \\) **do**
>         Update policy \\( \\{ \\theta_i \\}_{i=1}^n \\); Update value function \\( \\phi \\);
>     **end**
> **end**
>
> ---
>
> **Question1：**
> Is the use of the Fourier Transform for context length optimization novel in reinforcement learning? This is not clearly explained in the current version.
>
> **Answer：**
> In MARL, the input dimensionality grows exponentially with the number of agents, introducing substantial redundancy and noise into the feature space. Existing methods leveraging Fourier Transform (FT) in RL primarily focus on enhancing feature representation—for example, in state sequence modeling or market dynamics separation [5][6]—by preserving the full frequency spectrum. While this improves expressivity, it fails to suppress irrelevant or redundant components.
> In contrast, our work introduces a novel paradigm: strategic spectral truncation for context compression in discrete MARL. Rather than treating FT as a pure representational tool, we employ it as an information filtering mechanism, selectively retaining low-frequency components to capture long-term dependencies while discarding high-frequency noise.
>
> **Question2：**
> In Theorem~1, what is \\(\Omega\\), and why does \\( \\Omega(T) - O(T^{\\alpha}) = \\Omega(T) \\) when T is large?
>
> **Answer：**
> In the given expression, \\( \\Omega(T) \\) denotes an asymptotic lower bound, meaning there exist constants \\( c_1 > 0 \\) and \\( T_1 > 0 \\) such that \\( \\Omega(T) \\geq c_1 T \\) for all \\( T > T_1 \\); while \\( O(T^{\\alpha}) \\) represents an asymptotic upper bound, implying there exist constants \\( c_2 > 0 \\) and \\( T_2 > 0 \\) such that \\( O(T^{\\alpha}) \\leq c_2 T^{\\alpha} \\) for all \\( T > T_2 \\). When \\( T \\) is sufficiently large and \\( 0 \\leq \\alpha < 1 \\), \\( O(T^{\\alpha}) = o(T) \\) is a higher-order infinitesimal relative to \\( T \\), meaning \\( T^{\\alpha} \\) grows slower than \\( T \\) and \\( O(T^{\\alpha}) \\) becomes negligible compared to \\( \\Omega(T) \\) in the limit. Consequently, \\( \\Omega(T) - O(T^{\\alpha}) \\geq c_1 T - c_2 T^{\\alpha} \\), and since \\( c_2 T^{\\alpha} \\) is \\( o(T) \\), it follows that \\( c_1 T - c_2 T^{\\alpha} = \\Omega(T) \\) as the subtraction term does not alter the linear growth rate, leading to  \\( \\Omega(T) - O(T^{\\alpha}) = \\Omega(T) \\) for sufficiently large \\( T \\).
>
> **Question3：**
> Can the proposed method work in an offline learning setting?
>
> **Answer：**
> The proposed method, which includes Fourier low-frequency truncation and adaptive context optimization is inherently compatible with offline reinforcement learning. Firstly, low-frequency feature extraction operates deterministically on historical trajectories, requiring no interaction with the environment. Secondly, the context-length optimization module leverages temporal gradients from stored transitions, making it fully trainable in an offline setting. Moreover, our method is plug-and-play, independent of the specific RL backbone or whether learning is conducted in an online or offline regime.
>
> **Limitations Supplement**: As the number of agents increases, the central agent be subject to increased computational demands, which could impact overall processing efficiency. In future work, this may be addressed by adopting more advanced or specialized network architectures to improve scalability and parallelization. This discussion will be included in the appendix of the revised version.
>
> [1] llis, B., Cook, J., Moalla, S., Samvelyan, M., Sun, M., Mahajan, A., Foerster, J. N., & Whiteson, S. (2023). SMACv2: An improved benchmark for cooperative multi-agent reinforcement learning. NeurIPs 2023
>
> [2] Yu, C., Velu, A., Vinitsky, E., Gao, J., Wang, Y., Bayen, A., & Wu, Y. (2022). The surprising effectiveness of PPO in cooperative multi-agent games. NeurIPS 2022
>
> [3] Rashid, T., Samvelyan, M., Schroeder de Witt, C., Farquhar, G., Foerster, J., & Whiteson, S. (2018). QMIX: Monotonic value function factorisation for deep multi-agent reinforcement learning, ICML 2018
>
> [4] Wang, J., Ren, Z., Liu, T., Yu, Y., & Zhang, C. (2020). Qplex: Duplex dueling multi-agent q-learning. arXiv preprint arXiv:2008.01062.
>
> [5] Ye, M., Kuang, Y., Wang, J., Rui, Y., Zhou, W., Li, H., & Wu, F. (2023). State sequences prediction via fourier transform for representation learning. Advances in Neural Information Processing Systems, 36, 67565-67588.
>
> [6] Jeon, J., Park, J., Park, C., & Kang, U. (2024, August). Frequant: A reinforcement-learning based adaptive portfolio optimization with multi-frequency decomposition. In Proceedings of the 30th ACM SIGKDD Conference on Knowledge Discovery and Data Mining (pp. 1211-1221).

---

> ### Author Response · Authors · 2025-08-04
> **Additional Experiments On Large-Scale Tasks**
>
> We thank the reviewer for the valuable suggestion.
>
> We would like to clarify that the scenarios selected in our original submission—3s5z_vs_3s6z, 5m_vs_6m, and corridor—are among the relatively most challenging SMACv2 tasks, and have been widely adopted in prior works [1][2]. These tasks exhibit clear non-Markovian characteristics, such as delayed rewards, position-sensitive cooperation, and temporally extended dependencies—making them particularly suitable for assessing the strengths of ACL-LFT.
> To further demonstrate the scalability of our method, we have additionally conducted experiments on two large-scale scenarios with higher agent counts: MMM2 (10 agents) and 27m_vs_30m (27 agents). The results are summarized below.
> |SMACv2 (win rate)|RL-Method|Transformer|ToST|AMAGO|ACL-LFT|
> |:-:|:-:|:-:|:-:|:-:|:-:|
> |MMM2|MAPPO|92.4±2.5|92.9±3.1|94.1±2.9|95.5±2.3|
> | |QMIX|93.0± 2.2|93.6±2.4|94.7±1.8|95.9±1.9|
> | |QPLEX|93.2±2.3|93.3±2.6|94.9±2.8|96.2±2.4|
> |​​27m_vs_30m|MAPPO|90.5±3.9|91.3±3.4|93.2±3.6|94.6±3.3|
> | |QMIX|91.7±3.5|91.4±3.8|93.5±3.3|94.8±3.6|
> | |QPLEX|91.4 ± 3.7|92.6±4.0|94.1±3.8|95.1±3.5|
>
> [1] Li, C., Wang, T., Wu, C., Zhao, Q., Yang, J., & Zhang, C. (2021). Celebrating diversity in shared multi-agent reinforcement learning. Advances in Neural Information Processing Systems, 34, 3991-4002.
> [2] Rutherford, A., Ellis, B., Gallici, M., Cook, J., Lupu, A., Ingvarsson Juto, G., ... & Foerster, J. (2024). Jaxmarl: Multi-agent rl environments and algorithms in jax. Advances in Neural Information Processing Systems, 37, 50925-50951.

---

> > ### Comment · Reviewer_jva4 · 2025-08-05
> >
> > I thank the authors for the additional experiments, which are useful.

---

> > > ### Author Response · Authors · 2025-08-05
> > >
> > > Thank you very much for your positive and thoughtful comments. We will incorporate your suggestions into our next revision.

---

### Note · Authors · 2025-08-12

Dear Program Chairs, Senior Area Chairs, Area Chairs, and Reviewers,

We sincerely thank all reviewers for their insightful feedback and constructive discussions, which have significantly strengthened our submission. Through the rebuttal process, we clarified key methodological aspects (e.g., low-frequency truncation and Dec-POMDP framing), added comprehensive experiments on challenging benchmarks like SMACv2 (demonstrating consistent 3-5% gains across diverse MARL algorithms), and reinforced the method's efficiency, scalability, and adaptability to long-term dependency environments.

These enhancements highlight ACL-LFT's novel contributions in overcoming limitations of fixed context lengths in MARL—including limited exploration efficiency and redundant information—via a central agent for adaptive optimization and Fourier-based low-frequency truncation for effective redundancy filtering and global temporal trends extraction. We commit to incorporating all suggestions, including expanded limitations and clarifications, in the final version.

Thank you again for your time and valuable input.

Best regards,

The Authors

---

### Decision · Program_Chairs · 2025-09-17

**Decision:**

Accept (poster)

**Comment:**

This paper proposes adaptive context length optimisation with low-frequency truncation for MARL to handle long-range non-Markovian problems.

All the reviewers are positive about this paper. The only reviewer with a negative score also increased their score during the rebuttal. There were concerns about the scale of the environments, but the authors provided results on complex SMACv2 during the rebuttal. The authors properly addressed all the concerns, which resulted in an increase in scores by all reviewers.

As one of the reviewers mentioned, it is worth publishing this work because it pushes the state-of-the-art in long context MARL.